# Harnessing Temporal Databases for Systematic Evaluation of Factual Time-Sensitive Question-Answering in LLMs

**Soyeon Kim**[1*]**, Jindong Wang**[2]**, Xing Xie**[3]**, and Steven Euijong Whang**[1]

[1] KAIST, {`purplehibird,swhang`}`@kaist.ac.kr`
[2] William & Mary, `jdw@wm.edu`
[3] Microsoft Research Asia, `xing.xie@microsoft.com`

## Abstract

Facts change over time, making it essential for Large Language Models (LLMs) to handle time-sensitive factual knowledge accurately and reliably. Although factual Time-Sensitive Question-Answering (TSQA) tasks have been widely developed, existing benchmarks often face manual bottlenecks that limit scalable and comprehensive TSQA evaluation. To address this issue, we propose TDBench, a new benchmark that systematically constructs TSQA pairs by harnessing temporal databases and database techniques, such as temporal functional dependencies, temporal SQL, and temporal joins. We also introduce a new evaluation metric called time accuracy, which assesses the validity of time references in model explanations alongside traditional answer accuracy for a more fine-grained TSQA evaluation. Extensive experiments on contemporary LLMs show how TDBench enables scalable and comprehensive TSQA evaluation while reducing the reliance on human labor, complementing current TSQA evaluation approaches that largely center on Wikipedia/Wikidata by enabling LLM evaluation on application-specific data.

## 1 Introduction

Facts are not static – they evolve over time (Jensen et al., 1996). As Large Language Models (LLMs) are increasingly integrated into real-world applications (Jiao et al., 2024), their abilities to manage time-sensitive factual knowledge have become crucial for ensuring both accuracy and reliability (Yuan et al., 2024). For example, when asked, "Who is the current president of the U.S.?", an LLM must accurately distinguish between past and present presidents to avoid providing outdated or incorrect responses, which can mislead users and undermine trust (Huang et al., 2024).

To assess LLMs' abilities to handle time-sensitive factual knowledge, many Time-Sensitive Question Answering (TSQA) benchmarks have been developed (Chen et al., 2021; Dhingra et al., 2022; Kasai et al., 2023; Tan et al., 2023; Vu et al., 2023; Kim et al., 2024; Zhao et al., 2024; Zhu et al., 2025). These benchmarks primarily assess two LLM capabilities: (1) *temporal reasoning*, the ability to understand various temporal contexts within questions (e.g., "before 2019", "during the 5th Winter Olympics") and (2) *temporal alignment*, the ability to provide answers that reflect current factual knowledge in the real world (e.g., being aware of the current president).

However, existing factual TSQA benchmarks often lack a systematic design, relying heavily on manual efforts for benchmark construction and maintenance. Assessing temporal reasoning ability requires creating diverse temporal contexts such as "before 2019" or "during the 5th Winter Olympics", which typically relies on human writers (Kasai et al., 2023; Wei et al., 2023; Vu et al., 2023). Using predefined question templates can automate part of this process (e.g., "before [`YEAR`]"), but it still requires human efforts to design templates and often uses a small, fixed set that compromises diversity (e.g., 9 templates used in Dhingra et al. (2022), 16 used in Margatina et al. (2023); see more details in Sec. 5). In addition, the evolving nature of time-sensitive knowledge requires continual updates of TSQA benchmarks, posing a benchmark maintenance challenge. For instance, RealTimeQA (Kasai et al., 2023) manually curated weekly news (e.g., CNN) to incorporate new world facts, but updates have recently ceased due to the high costs of manual curation (Uddin et al., 2024).

---

*Work done during an internship at Microsoft Research Asia.

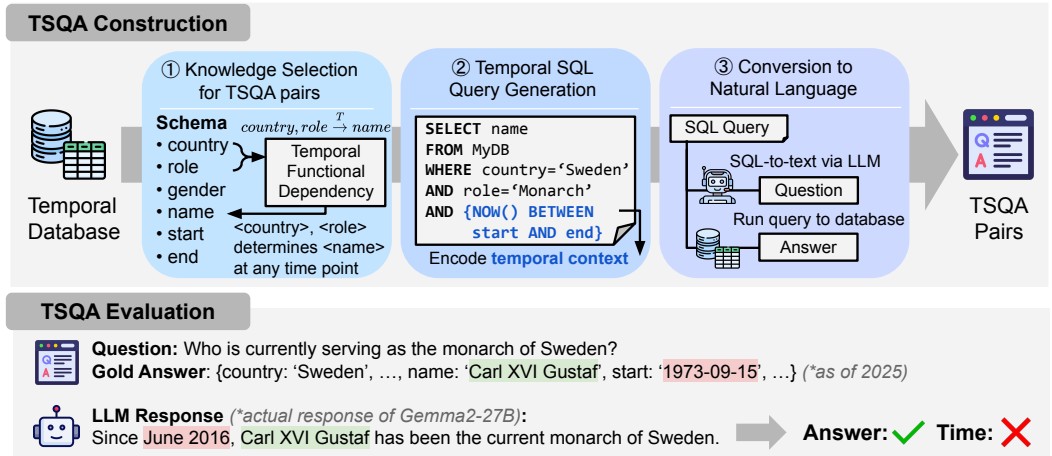

Figure 1: **Overview of TDBench framework.** TDBench systematically constructs Time-Sensitive QA (TSQA) pairs by (1) selecting factual knowledge via temporal functional dependencies, (2) generating temporal SQL queries with diverse temporal contexts, and (3) converting queries into natural language QA pairs using an LLM and the database. During evaluation, TDBench automatically verifies both the final answer and time references in LLM responses, capturing cases where the model hallucinates in the explanation despite providing the correct answer. TDBench supports diverse TSQA scenarios, including temporal alignment, temporal reasoning, and implicit multi-hop questions.

To tackle these manual bottlenecks, we propose **TDBench**, a benchmarking framework designed for a more systematic TSQA evaluation. Unlike existing TSQA benchmarks, TDBench eliminates the need for human labor in designing temporal contexts or question templates. Instead, TSQA pairs are automatically generated from an input data source, namely, *temporal databases* – extensions of conventional databases that are well-established for managing time-sensitive knowledge (Jensen et al., 1996). These temporal databases can be user-defined or sourced from general platforms like Wikipedia, allowing users to flexibly construct TSQA benchmarks based on their own interests.

Our key idea is to utilize database techniques for automatic TSQA construction, namely, Temporal Functional Dependencies (TFDs), temporal SQL, and temporal joins. Fig. 1 illustrates TDBench's 3-step process: (1) selecting factual knowledge suitable for TSQA tasks by leveraging TFDs, which pinpoint attributes whose values can be uniquely determined at any timepoint (e.g., a country's president at a given time); (2) generating temporal SQL queries to encode diverse temporal contexts, leveraging Allen's interval algebra (Allen, 1983) to cover 13 mutually exclusive and exhaustive temporal relations via rich temporal SQL operators (e.g., `BETWEEN`, `DATEDIFF`); and (3) translating SQL queries into natural language QA pairs using an LLM and the database. Through these steps, TDBench simplifies TSQA construction and maintenance; updating the database content with temporal changes (e.g., new presidents) automatically refreshes the corresponding QA pairs. We also show how temporal joins increase TSQA complexity by generating implicit temporal contexts (e.g., "during the 5th Winter Olympics"), which can test LLMs' advanced reasoning abilities such as event–event reasoning (Tan et al., 2023) without typically required manual curation (Tan et al., 2023).

Alongside the QA construction, we introduce a new metric called *time accuracy* to support a more fine-grained TSQA evaluation. During QA tasks, LLMs can often hallucinate in explanations while providing correct answers (Ji et al., 2023; Oh et al., 2024). We find this phenomenon manifests in TSQA tasks as inaccurate time references, as illustrated in Fig. 1; a model correctly identifies the current monarch of Sweden, but hallucinates his start date. Since traditional answer-only evaluations cannot capture such errors, we use time accuracy to verify the validity of time references as well as final answers, enabling automatic verification with SQL-based temporal constraints.

Extensive experiments demonstrate how TDBench enables a scalable and comprehensive TSQA evaluation while reducing human labor. We assess several popular LLMs – GPT-3.5 (OpenAI, 2022), GPT-4 (OpenAI, 2023), GPT-4o (OpenAI, 2024), Llama3.1-70B (Dubey et al., 2024), Mixtral-8x7B (Jiang et al., 2024), Gemma2-27B (Team et al., 2024), Qwen2-72B (Bai et al., 2023), and Granite3.1-8B (Granite Team, 2024) – across various TSQA tasks, including temporal alignment, temporal reasoning, and multi-hop settings with implicit temporal contexts.

Our benchmark offers several key distinctions and findings. First, TDBench moves beyond previous Wikipedia/Wikidata-centric TSQA evaluation by leveraging domain-specific databases (e.g., medical, legal) not covered by public platforms. Second, our time accuracy metric reveals that LLMs largely hallucinate time references despite providing correct answers (on average 21.7% in our experiments), uncovering a reliability issue of LLMs in TSQA. Third, TDBench's SQL-generated temporal contexts encompass 13 diverse constraint types compared to 4-6 in other TSQA benchmarks, identifying LLMs' temporal blind spots that are not clearly captured by existing benchmarks. Code and data are publicly available at: https://github.com/ssoy0701/tdbench.git.

**Summary of Contributions**  (1) We propose TDBench, a new TSQA benchmark that systematically constructs QA pairs without human labor by harnessing temporal databases and database techniques. (2) We introduce a time accuracy metric to assess the validity of temporal references in model explanations, enabling more reliable TSQA evaluation beyond answer correctness alone. (3) Extensive experiments demonstrate that TDBench provides comprehensive TSQA evaluation through diverse SQL-generated temporal questions and complements prior Wikipedia/Wikidata-centric benchmarks.

## 2   BACKGROUNDS

**Temporal Database**  Temporal databases (Jensen & Snodgrass, 2018) extend conventional relational databases with the concept of time. Both types of databases store structured data, where information is organized into a fixed schema with predefined attributes (i.e., columns) and corresponding values (i.e., rows or tuples). While conventional relational

Table 1: **An example temporal database** *Leader(country, role, gender, name, start, end).*

| country | role | gender | name | start | end |
|---|---|---|---|---|---|
| USA | President | M | Bush | 2001 | 2009 |
| USA | President | M | Obama | 2009 | 2017 |
| U.K. | Monarch | F | Elizabeth | 1952 | 2022 |

databases typically retain only the latest state of each entity, temporal databases are specifically designed to capture time-varying information by storing historical states with associated timestamp attributes. In this work, we focus on temporal databases following the uni-temporal data model, which stores the validity interval of each row using two timestamp attributes (e.g., *start* and *end* in Table 1). These temporal databases are widespread across many critical domains (see Sec. A.1), and we show how the structured nature of these data enables systematic TSQA benchmarking.

**Temporal Functional Dependency and Join**  Temporal databases possess two unique properties: 1) *temporal functional dependencies* (TFDs) and 2) *temporal joins*. TFDs adapts the notion of Functional Dependencies (FDs) from conventional databases to the time dimension [1]. While an FD $X \rightarrow Y$ requires the values of $X$ to uniquely determine the values of $Y$ across the entire relation, a TFD $X \xrightarrow{T} Y$ requires this dependency to hold whenever tuples are simultaneously valid in time, i.e., when their validity intervals overlap (Jensen et al., 1996). For example, the dependency *country, role* $\xrightarrow{T}$ *name* in Table 1 ensures that, during any time period, a given role within a country is held by at most one individual, while allowing the role-holder to change over time. In contrast, a standard FD would require the same role and country to always map to the same person, disallowing any change over time. This highlights how TFDs generalize FDs by capturing temporal variation. Similarly, temporal joins incorporate time when combining two or more tables. We mainly focus on temporal natural joins, which only join tuples with overlapping validity intervals. In the following sections, we leverage these properties to automatically construct QA pairs without manual effort. Formal definitions of both properties are presented in Sec. A.2.

## 3   TDBENCH

We introduce TDBench, a new benchmarking framework for systematic evaluation of Time-Sensitive Question Answering (TSQA). We first describe the automatic construction of TSQA pairs from temporal databases, which greatly reduces reliance on manual effort in constructing and updating QA pairs (Sec. 3.1). We then outline an evaluation protocol that moves beyond answer-only evaluation and assesses models' temporal reasoning and explanations (Sec. 3.2). Finally, we discuss an extension that increases TSQA difficulty by incorporating implicit temporal contexts (Sec. 3.3).

---

[1]Both FDs and TFDs are defined by the database owner. These properties are typically imposed in databases to detect data inconsistencies and enforce data integrity (Jensen et al., 1996).

Table 2: **Examples of temporal context generation using Allen's interval algebra.** Interval relations (e.g., *before, meet, overlap*) are implemented with SQL conditions to generate temporal contexts, where $a$ is a tuple's validity interval (e.g., a president's term) and $b$ is a randomly sampled time interval (e.g., ['2000-09-20', '2001-03-20']) to generate diverse temporal contexts. The overlap relation has two implementations: one for general temporal reasoning and one for modeling the "current" condition used in the temporal alignment task. See Table 10 for all 13 temporal relations.

| Relation | Interval Diagram | SQL Condition | Example Temporal Context |
|---|---|---|---|
| $a$ before $b$ | | $a$.end < $b$.start | A president who ends before *September 20, 2000* |
| $a$ meet $b$ | | $a$.end = $b$.end - $b$.length | A president who ends exactly *half a year* before *March 20, 2001* |
| $a$ overlap $b$ | | $a$.start < $b$.start $\wedge$ b.start < a.end < b.end | A president who starts before *September 20, 2000* and ends between *September 20, 2000* and *March 20, 2001* |
| | | $a$.start < $b$.start $\wedge$ $a$.end IS NULL | A president who is currently serving |

## 3.1 SYSTEMATIC QA CONSTRUCTION

Our key idea is to harness temporal database techniques to systematize the QA construction process. As shown in Fig. 1, TDBench proceeds in the following three steps, yielding several benefits.

① **Knowledge Selection via TFDs**  Given a temporal database, we use TFDs (Sec. 2) to automatically select attribute sets for constructing TSQA pairs. Given a TFD $X \xrightarrow{T} Y$ satisfied in a database relation $r$, we specify $r.X$ values in the question and use the corresponding $r.Y$ values as the answers, since $X$ values determine $Y$ values at any time by the TFD definition. For example, $country, role \xrightarrow{T} name$ leads to the question "Who is the [role] of the country [country]"? with the answer [name], where [role], [country], and [name] are placeholders for attribute values. As TFDs are part of the design theory of temporal databases, this FD-based logic generalizes to arbitrary schemas of temporal databases, enabling systematic knowledge selection. While TFDs are commonly defined in temporal databases, we discuss how to handle cases where TFDs are missing, unknown, or invalid in Sec. B.3.

② **Temporal SQL Query Generation**  Using the TFD-selected attributes, we generate temporal SQL queries to cover diverse temporal questions. SQL stands for Structured Query Language, which is used to interact with databases to manage and query data. Rather than directly generating natural language questions, generating questions via temporal SQL queries allows us to utilize expressive built-in temporal operators, reducing the manual effort to design temporal contexts.

We design `Genqueries` algorithm to automatically generate SQL queries. `Genqueries` first (1) builds a base query using the TFD-selected attributes and then (2) adds temporal constraints, as shown in Fig. 1. For (1), the $X$ attributes are used in the `WHERE` clause for the question, and the $Y$ attributes are used in the `SELECT` clause for the answer (e.g., $country, role \xrightarrow{T} name$ yields `SELECT name WHERE country='USA' AND role='president'`). For (2), SQL conditions are generated based on Allen's interval algebra (Allen, 1983), a well-defined framework from the database literature that specifies 13 mutually exclusive and exhaustive temporal relations: *before, after, meet, met-by, overlap, overlapped-by, equal, start, started-by, finish, finished-by, during*, and *contain*. For example, Table 2 shows the temporal SQL condition and the temporal context corresponding to the *meet* relation, where a time interval $a$ ends exactly when a time interval $b$ starts. The resulting temporal condition is appended to the base query as a temporal constraint. We present the pseudocode of `Genqueries` and the full set of SQL conditions in Sec. B.1 and Sec. B.2, respectively.

③ **Conversion to Natural Language QA Pairs**  We convert the generated queries into natural language QA pairs by using an LLM and the underlying database. For questions, we use GPT-4o (OpenAI, 2024) as an SQL-to-text translator via system prompts, which we observe to achieve 91.5% accuracy in our setup based on LLMs' strong zero-shot performance in SQL-to-text tasks (Zhang et al., 2024a) – see detailed setups (e.g., actual prompt and temperature) and error analyses in Sec. B.4. For answers, we simply run the generated SQL queries on the database. In this way, we obtain

Table 3: **QA pair generation from an SQL query.** Each query is translated to multiple natural language questions with the same answer. To evaluate both answer and time accuracy, the query also outputs a relation-specific time reference (e.g., `end` for the *meet* relation), defined in Table 10.

| SQL Query (relation='meet') | Generated QA |
|---|---|
| `SELECT name, end FROM Leader`
`WHERE Country='Brazil' AND`
`Role='President' AND`
`date(end) = date('2019-05-01',`
`'-4 month')` | **[Questions]** "Who was the president of Brazil whose term ended exactly four months before May 1, 2019?", "Can you provide the president of Brazil who finished their term four months prior to May 1, 2019?"...
**[Answer]** Michel Temer   **[Time reference]** 2019-01-01 (end) |
| **[Model Response]** The answer is *Michel Temer* (answer), whose term ended in *January 1, 2019* (time reference). | |

linguistically diverse questions from an LLM while ensuring that the answers follow the underlying database, as shown in Table 3. We show more examples of generated QA pairs in Sec. B.4.

**Benefits**   Compared to prior approaches, TDBench offers many benefits by harnessing both database techniques and LLMs for QA construction: (1) *convenient and generalizable benchmarking.* TFDs eliminates the need for manual data pre-processing to identify time-related attributes (Gupta et al., 2023; Zhao et al., 2024) while generalizing to any temporal databases; (2) *automatic QA construction with rich temporal constraints.* Temporal SQL eliminates the need for customized, human-written templates while expanding temporal constraints to 13 types, instead of the typically used 4-6 types (mainly 'in', 'from-to', 'before', and 'after' (Chen et al., 2021; Dhingra et al., 2022)); (3) *better accuracy and cost efficiency compared to LLM-only approaches.* By grounding answers in the database, we demonstrate TDBench achieves lower QA pair error rates and reduces LLM inference costs compared to relying only on LLMs for QA construction (Kim et al., 2024) (Sec. D.8).

## 3.2   EVALUATION OF BOTH ANSWERS AND TIME REFERENCES

Using the generated QA pairs, TDBench evaluates not only final answers, but also explanations for a more reliable TSQA evaluation. To this end, we propose *time accuracy* metric and explain its definition and verification process, along with the evaluation scenarios supported by TDBench.

**Definition of Time Accuracy**   We define time accuracy as the correctness of time references – specifically the start and end dates in our setup – that serve as rationales for temporal reasoning required in a time-sensitive question. For example, the questions in Table 3 require reasoning over the president's end date, where the model should mention "*January 1, 2019*" in the explanation given the constraint "*four months before May 1, 2019*". If a question requires both a start and an end date, but the model gets only one correct, the time accuracy is 50%. See the formal definition in Sec. B.5.

**Verification of Time Accuracy**   TDBench supports automatic verification of time accuracy by utilizing the SQL-generated temporal constraints, which explicitly encode the temporal reasoning logic (e.g., "date(end) = date('2019-05-01'), -4 month" in Table 3 indicates that the end date is required for reasoning). We thus define relation-specific criteria that specifies which time references to verify: start date for '*after*', '*met-by*', '*started-by*', end date for '*before*', '*meet*', '*finished-by*', and both dates for the remaining seven relations, as summarized in Table 10. To facilitate evaluation, we prompt target LLMs to explicitly state time references used in their reasoning during the TSQA task. Since model responses may include unrelated time information (e.g., "As of 2025, the answer is..."), we employ an LLM-based judge rather than exact matching to verify time references, achieving 91.1% accuracy under manual inspection – see more details and error analyses in Sec. B.6. We also demonstrate how the time accuracy metric can be supported in other TSQA benchmarks in Sec. 4.1.

**Evaluation Scenarios**   TDBench supports various TSQA evaluation scenarios: (1) *temporal alignment evaluation*, which tests whether the LLM is up-to-date with current world knowledge. As this scenario grounds questions to a specific temporal constraint "current", we use the '*overlap*' relation to generate the "current" constraint (Sec. B.2); (2) *temporal reasoning evaluation*, which assesses the LLM's understanding of diverse temporal contexts, where we can use all 13 types of temporal constraints generated by `Genqueries`; (3) *open-book/closed-book evaluation*, which differs by the presence of additional context that can aid the QA task. In the open-book setting, we append both relevant and irrelevant rows from the database to the question, whereas such context is removed in the closed-book setting (see Sec. B.7 for an example QA in each setting). We note

that these scenarios are enabled by the expressiveness of temporal SQL, which also allows natural extensions to broader scenarios – see how to extend `Genqueries` to incorporate non-temporal data attributes, semi-structured data, and other temporal tasks such as event-counting in Sec. B.1.

## 3.3 EXTENSION WITH IMPLICIT TEMPORAL CONTEXTS

To properly evaluate ever-improving LLMs, we can further increase question complexity by generating implicit temporal questions that require reasoning between multiple events. Consider the question $q$: "*Who was the president of the host country during the 1988 Summer Olympics?*". The temporal constraint here is implicit, expressed as "*during the 1988 Summer Olympics*" rather than explicit dates. These questions require an advanced temporal reasoning ability (Tan et al., 2023), where the model must correctly identify two temporally overlapping events.

TDBench generates such implicit questions like $q$ by performing temporal joins (Sec. 2) of tables within the SQL queries. For example, joining *Olympic(country, city, game_edition, start, end)* and *Leaders(country, role, gender, name, start, end)* on the common column '*country*' pairs Olympic host countries with their national leaders at that time. By combining information from both events through this join, the resulting table contains tuples suitable for generating questions that span multiple events (like $q$) and their corresponding answers. To automatically generate QA pairs, `Genqueries` uses TFDs from this joined table – which can be derived via standard FD and TFD inference rules (Jensen et al., 1996) – while skipping the explicit temporal constraint generation process to create implicit constraints instead. For instance, $game\_edition \rightarrow country$ (in *Olympic*) and $country, role \xrightarrow{T} name$ (in *Leaders*) yield $game\_edition, role \xrightarrow{T} name$, which `Genqueries` converts to "Who was the [role] of the host country during the [game edition]?" The model response evaluation process remains the same as for questions with explicit temporal constraints.

## 4 EXPERIMENTS

We perform extensive experiments to demonstrate how TDBench enables diverse and fine-grained temporal evaluation of LLMs. More detailed settings (e.g., system prompts) are provided in Sec. C.

**LLMs Compared**  We compare GPT-3.5 (OpenAI, 2022), GPT-4 (OpenAI, 2023), GPT-4o (OpenAI, 2024), Llama3.1-70B (Dubey et al., 2024), Mixtral-8x7B (Jiang et al., 2024), Gemma2-27B (Team et al., 2024), Qwen2-72B (Bai et al., 2023), and Granite3.1-8B (Granite Team, 2024). We set all the temperature parameters to zero to exclude randomness in the LLM responses.

**Datasets**  We use temporal databases from two types of sources: (1) Wikipedia covering domains such as *Countries*, *Athletes*, *Organizations*, and *Olympics*, which are commonly addressed in existing TSQA benchmarks (Chen et al., 2021; Dhingra et al., 2022; Mousavi et al., 2024), generating 6,177 questions; and (2) other platforms such as Kaggle, covering areas including same-sex marriage laws, carbon taxation, UNESCO heritage, and Netflix shows to model domain-specific scenarios in *Legal*, *Environmental*, *Cultural*, and *Social* contexts, generating 1,704 questions. More details on datasets, including schemas, used TFDs, and subcategories of questions, are provided in Sec. C.2.

**Performance Measures**  We mainly use the following three performance evaluation metrics.
- *Answer accuracy* (**A**): Portion of responses with correct answers.
- *Time accuracy* (**T**): Portion of responses with correct time references, evaluated using relation-specific criteria (Table 10). If two time references are expected (e.g., start and end dates), partial credit (50%) is given when only one is correct. Granularity (e.g., year, month) varies by dataset.
- *Answer-Time accuracy* (**AT**): Portion of responses with both correct answers and time references. Here, we do not give partial credit when assessing time references (i.e., only consider 100%), providing a strict metric for more rigorous analysis.

## 4.1 EVALUATION OF TEMPORAL ALIGNMENT

We evaluate the temporal alignment of LLMs, assessing the ability to answer up-to-date questions that reflect current world knowledge (Dhingra et al., 2022; Kasai et al., 2023; Mousavi et al., 2024).

**Performance Variations when Accounting for Time Accuracy**  Table 4 shows notable performance variations across domains under two evaluation settings: **A** (answer-only) and **AT** (answer and time).

Table 4: **Performance evaluation of eight LLMs on the temporal alignment TSQA task.** We report the proportion of correct responses under two evaluation settings: answer-only ($\mathbf{A}$) and answer and time ($\mathbf{AT}$), along with their difference ($\Delta = \mathbf{A} - \mathbf{AT}$). Wikipedia results are reported in aggregate, and domain-specific results are broken down by subdomain. Best results are shown in bold.

| Model | Wikipedia | | | Law | | | Carbon Tax | | | Heritage | | | Netflix | | |
|---|---|---|---|---|---|---|---|---|---|---|---|---|---|---|---|
| | A | AT | $\Delta$ | A | AT | $\Delta$ | A | AT | $\Delta$ | A | AT | $\Delta$ | A | AT | $\Delta$ |
| GPT-3.5 | 47.5 | 22.4 | 25.1 | 84.3 | 39.9 | 44.3 | 19.9 | 8.4 | 11.4 | 62.7 | 26.5 | 36.2 | 29.3 | 19.1 | 10.1 |
| GPT-4 | 44.5 | 29.6 | 14.9 | 82.4 | 52.3 | **30.0** | 32.0 | 23.9 | **8.1** | 74.5 | 44.5 | 30.0 | 32.4 | 26.4 | 6.1 |
| GPT-4o | **73.9** | 48.8 | 25.1 | 84.3 | **54.0** | 30.3 | 72.1 | **43.1** | 29.0 | 72.5 | **53.3** | 19.2 | 32.9 | 25.9 | 7.0 |
| Llama3.1-70B | 64.6 | **56.7** | **7.9** | 84.6 | 44.9 | 39.7 | 57.6 | 39.7 | 17.8 | **75.5** | 39.7 | 35.8 | 35.6 | **28.6** | 7.0 |
| Mixtral-8x7B | 42.9 | 30.9 | 12.0 | 74.1 | 42.2 | 32.0 | 28.3 | 16.5 | 11.8 | 35.6 | 18.1 | **17.5** | 20.7 | 17.1 | 3.6 |
| Gemma2-27B | 69.3 | 32.8 | 36.5 | 84.3 | 29.8 | 54.6 | **73.1** | 25.2 | 47.8 | 69.2 | 22.0 | 47.2 | **36.0** | 27.7 | 8.3 |
| Qwen2-72B | 62.7 | 34.1 | 28.6 | 84.0 | 34.2 | 49.9 | 57.2 | 27.6 | 29.6 | 53.5 | 20.2 | 33.3 | 22.5 | 19.4 | **3.1** |
| Granite3.1-8B | 49.6 | 26.1 | 23.5 | **90.1** | 28.6 | 61.4 | 45.1 | 0.0 | 45.1 | 42.0 | 4.5 | 37.5 | 19.6 | 14.9 | 4.7 |
| **Average** | 56.9 | 35.2 | 21.7 | 83.5 | 40.7 | 42.8 | 48.2 | 23.1 | 25.1 | 60.7 | 28.6 | 32.1 | 28.6 | 22.4 | 6.2 |

While the models generally achieve high $\mathbf{A}$ by retrieving correct up-to-date answers, they often fail to generate accurate time references, resulting in a 21.7% average performance drop in $\mathbf{AT}$ on the Wikipedia dataset. This gap reveals the limitation of traditional answer-only evaluation, indicating that a large portion of factually inconsistent responses with incorrect time references can be overlooked – see actual cases in Sec. D.1. We also observe domain-specific LLM performances; GPT-4o performs best in domains such as law, carbon tax, and UNESCO heritage, while Llama3.1-70B excels in the Netflix domain – demonstrating TDBench's applicability beyond Wikipedia-based data.

**Integrating Time Accuracy into Existing Benchmarks** We show how the proposed time accuracy metric can be beneficial when integrated into existing temporal alignment TSQA benchmarks (Dhingra et al., 2022; Margatina et al., 2023; Mousavi et al., 2024), enabling more fine-grained evaluation. These benchmarks use open-ended QA templates (e.g., "The president of Italy is __") and solely evaluate the final answer. To integrate $\mathbf{T}$ and assess model explanation as well, we simply modify the system prompt to elicit start date references from the model, without altering the original benchmark. In Sec. D.2, we provide a detailed case study with Dyknow (Mousavi et al., 2024), where we extend the benchmark with $\mathbf{T}$ and observe an F1-score of 0.96 for temporally aligned responses (i.e., both answer and time reference are correct), improving benchmark correctness against human verifications.

## 4.2 BREAKDOWN OF TEMPORAL REASONING PERFORMANCE

We now evaluate the temporal reasoning ability of LLMs to understand diverse temporal conditions, analyzing performance by answer cardinality, temporal constraint type, and temporal span.

**Performance by Answer Cardinality** We evaluate LLM performances across different answer set sizes, expanding the traditional setup with a single answer. Depending on the generated temporal context, TDBench can yield multiple-answer or no-answer questions as well as single-answer questions. For example, a no-answer question might ask, *"Which president of the U.S. ended their term exactly half a year before January 20, 2001?"*, where no such president exists. Table 5 shows that models like GPT-4o perform robustly across question types, whereas models like Mixtral and GPT-3.5 show notable gaps, particularly underperforming on no-answer questions. For example, a model incorrectly answers "Bill Clinton" (term ended on January 20, 2001), likely due to matching the string "January 20, 2001" in the question rather than reasoning over "exactly half a year.", indicating that models may rely more on surface cues than true temporal reasoning.

**Performance by Temporal Constraint Type** We assess how well the LLMs handle different types of temporal constraints using the 13 distinct temporal relations. Fig. 2 shows that LLMs perform best on 'equal' relation-based constraints (e.g., "in 2025") – likely due to their prevalence in training data – while struggling with 'contain' and 'overlap', which require precise temporal reasoning over both start and end dates (e.g., started before January 1, 1985, and ended after December 30, 1992 for 'contain'). Interestingly, LLMs tend to comprehend start constraints (e.g., 'start', 'started-by') more accurately than end constraints (e.g., 'finish', 'finished-by'), which may stem from the autoregressive nature of LLMs that prioritizes earlier tokens in generation.

Table 5: **LLM performances by answer cardinality.** Performances are reported across multiple-, unique-, and no-answer ("None") question types, evaluated on the Wikipedia dataset in the open-book setting. Only **A** is reported for the None type questions, as no time reference is expected in no-answer cases.

| | Multiple | | | Unique | | | None |
|---|---|---|---|---|---|---|---|
| **Model** | **A** | **T** | **AT** | **A** | **T** | **AT** | **A** |
| GPT-3.5 | 34.4 | 41.9 | 20.1 | 43.3 | 67.8 | 36.8 | 17.9 |
| GPT-4 | **60.3** | **77.8** | **49.8** | 61.7 | 82.9 | 49.9 | 45.0 |
| GPT-4o | 58.1 | 67.3 | 42.2 | 67.8 | 68.7 | 43.7 | **62.4** |
| Llama3.1-70B | 48.6 | 45.7 | 29.4 | **69.1** | 51.8 | 33.2 | 44.2 |
| Mixtral-8x7B | 36.2 | 57.6 | 24.1 | 49.9 | 78.1 | 36.6 | 7.8 |
| Gemma2-27B | 49.7 | 56.1 | 29.1 | 61.9 | 63.6 | 41.7 | 20.0 |
| Qwen2-72B | 48.3 | 62.5 | 37.7 | 63.4 | **86.4** | **57.0** | 49.1 |
| Granite3.1-8B | 22.5 | 41.9 | 14.8 | 46.9 | 51.2 | 25.8 | 25.1 |

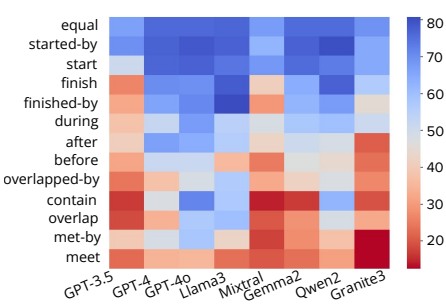

Figure 2: **LLM performances on 13 temporal relations in the open-book setting.** The heatmap displays **AT** for each temporal relation, as defined in Table 10.

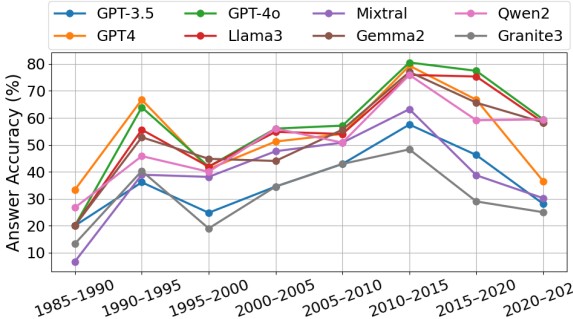

Figure 3: **LLM performances across different time spans (1985-2025) in a single domain (*Countries*) under the closed-book setting.** Additional results aggregated over multiple domains are presented in Sec. D.3.

Table 6: **LLM Performances on implicit, multi-hop questions (Sec. 3.3).** $H_i$ denotes the hallucination rate at the $(i + 1)$-th hop given the correct $i$-th hop. Full results with 2-hop questions are shown in Table 29.

| | 3-hop | | | |
|---|---|---|---|---|
| **Model** | **A** | **AT** | **H$_1$** | **H$_2$** |
| GPT-3.5 | 33.8 | 33.8 | **8.5** | 7.6 |
| GPT-4 | 57.1 | 52.5 | 10.8 | 0.5 |
| GPT-4o | **90.4** | **87.9** | 26.3 | **0.0** |
| Llama3.1-70B | 82.8 | 80.8 | 38.2 | 4.5 |
| Mixtral-8x7B | 60.1 | 49.0 | 19.5 | 10.6 |
| Gemma2-27B | 48.0 | 43.9 | 26.5 | 24.7 |
| Qwen2-72B | 68.2 | 59.1 | 35.1 | 5.6 |
| Granite3.1-8B | 11.1 | 11.1 | 10.3 | 77.8 |

**Performance by Temporal Span**   We evaluate LLM performance across different temporal spans under the closed-book setting, assessing how well LLMs utilize internal time-sensitive factual knowledge without external context. Fig. 3 shows the results for a single domain (*Countries*) (see Sec. D.3. for aggregated results across multiple domains) where we observe a gradual performance increase from 1985 to 2010, with a notable jump in the 1990–1995 interval. As this trend appears consistently across all models, we suggest it reflects shared characteristics of training data rather than model-specific behavior, where the 1990–1995 period may be better represented in the training data.

## 4.3 EVALUATION OF IMPLICIT, MULTI-HOP QUESTIONS

We assess LLM performance on multi-hop questions. As described in Sec. 3.3, these questions have implicit temporal contexts generated via temporal joins, and we construct 2-hop and 3-hop questions by joining the *Olympic* and *Country* datasets. For the 2-hop setting, we use the TFD $game\_edition, role \xrightarrow{T} name$, where models should infer (1) the host country and time from a given Olympic game edition (e.g., 1988 Summer Olympics), and (2) the leader of the host country (e.g., president) at that time. For the 3-hop setting, we use $game\_round, role \xrightarrow{T} name$, where models should infer (1) the Olympic game edition from the round number (e.g., 24th Summer Olympics), (2) the host country and time, and (3) the leader at that time. An example QA is shown in Sec. B.4.

**Performance by Reasoning Hops**   To effectively assess the reasoning process, we introduce $H_i$, which measures the conditional probability of hallucination where a model answers the $i$-th hop correctly, but hallucinates on the $(i+1)$-st hop. Alongside the overall performances captured by **A** and **AT**, Table 6 shows the step-wise performances captured by $H_i$, revealing the most challenging hop for each model. For example, GPT-4o and Llama3.1-70B hallucinate more at the first hop (i.e.,

higher $H_1$), while Granite3.1-8B hallucinates more at the second hop (i.e., higher $H_2$) in the 3-hop setting. These model-specific challenges highlight the value of TDBench's multi-hop generation capability, which enables a fine-grained diagnosis of intermediate reasoning failures.

**Impact of Performance Improvement Techniques** To mitigate reasoning errors in multi-hop settings, we evaluate the effectiveness of prompting strategies known to support model reasoning. Specifically, we compare RAG (Lewis et al., 2020), which augments prompts with retrieved Wikipedia passages; Chain-of-Thought (CoT) (Wei et al., 2022), which guides models to generate intermediate reasoning steps; and Time-CoT (Yin & Hu, 2024), a recent CoT variant designed for temporal reasoning – see CoT and Time-CoT prompts in Sec. D.4. As shown in Table 7, CoT and Time-CoT generally outperform RAG, which often retrieves relevant yet temporally misaligned contexts (see a case study in Sec. D.4). Between CoT

Table 7: **Performance comparison on multi-hop questions when applying RAG, CoT, and Time-CoT.** Best- and second-best performing methods for each model are highlighted in bold and underline, respectively.

| Model | Original | RAG | CoT | Time-CoT |
|---|---|---|---|---|
| GPT-3.5 | 63.1 | **71.2** | 39.4 | 15.2 |
| GPT-4 | 67.2 | 35.4 | 92.4 | **93.9** |
| GPT-4o | 79.3 | 89.9 | 92.9 | **95.5** |
| Llama3.1-70B | 83.8 | 73.2 | **92.9** | 91.9 |
| Mixtral-8x7B | **72.7** | 60.1 | 67.2 | 71.2 |
| Gemma2-27B | 77.8 | 40.9 | **83.8** | 78.8 |
| Qwen2-72B | 65.2 | 78.8 | 77.8 | **85.4** |
| Granite3.1-8B | 50.5 | 54.5 | 70.2 | **72.2** |

and Time-CoT, no single method consistently outperforms the other; see results with single-hop questions and additional analysis of RAG's underperformance in Sec. D.4.

## 4.4 CORRECTNESS OF TDBENCH

We evaluate the correctness of TDBench across three task types (Sec. 4.1 – Sec. 4.3). We compare TDBench's evaluation results with manual verification on 125 randomly sampled responses per task, using precision (P), recall (R), and F1-score (F1); see Sec. D.5 for metric definitions and details of manual verification. As shown in Table 8, TDBench achieves high agreement with manual verification, demonstrating the effectiveness of database-driven techniques for LLM benchmarking.

Table 8: **Correctness of TDBench against manual verification.** Fine-grained results are in Table 24.

| Methods | P | R | F1 |
|---|---|---|---|
| Temporal alignment | 0.98 | 0.96 | 0.97 |
| Temporal reasoning | 0.87 | 0.91 | 0.88 |
| Multi-hop | 0.95 | 0.87 | 0.91 |

## 4.5 MORE ANALYSES

To further evaluate the applicability and robustness of TDBench, we conduct additional analyses: (1) *evaluation on synthetic medical data*, demonstrating how TDBench can be applied beyond real-world facts (Sec. D.6); (2) *evaluation of task-specific TSQA methods*, which underperform relative to the general-purpose models in the main experiments (Sec. D.7); (3) *comparison with LLM-based TSQA methods*, which generate TSQA pairs by directly feeding raw data into LLMs, but can be more susceptible to hallucination and incur higher inference costs when processing the raw data (Sec. D.8); and (4) *cross-translator similarity analysis*, which examines the impact of different SQL-to-text translators on question phrasing (Sec. D.9).

## 5 RELATED WORK

**Factual Time-Sensitive Question-Answering** TDBench differs from prior factual TSQA benchmarks in three main aspects: data source, QA construction, and evaluation methodology. Prior TSQA studies heavily centers on Wikidata (Jia et al., 2018; Chen et al., 2021; Dhingra et al., 2022; Margatina et al., 2023; Tan et al., 2023; Mousavi et al., 2024; Luo et al., 2025; Zhu et al., 2025; Islakoglu & Kalo, 2025) or Wikipedia (Jang et al., 2022; Kim et al., 2024; Wu et al., 2024; Xiong et al., 2024; Gruber et al., 2025). While some recent work target tabular data (Gupta et al., 2023; Zhao et al., 2024), they remain limited to Wikipedia tables and often require

Table 9: **Comparison with TDBench against template-based factual TSQA benchmarks.** We cover 13 temporal relations (Op #), without relying on the fixed set of templates (Temp #) while verifying both answers (**A**) and time references (**T**) during evaluation (Eval).

| Method | Op # | Temp # | Eval |
|---|---|---|---|
| Dhingra et al. (2022) | 1 (Equal) | 9 | **A** |
| Margatina et al. (2023) | 1 (Equal) | 16 | **A** |
| Tan et al. (2023) | 6 | 23 | **A** |
| Mousavi et al. (2024) | 1 (Equal) | 27 | **A** |
| TDBench (Ours) | 13 | (Unlimited) | **A, T** |

additional Wikipedia page content for QA construction, restricting generalization beyond Wikipedia. In contrast, TDBench is not constrained by Wikidata/Wikipedia domains, enabling generalization

to arbitrary temporal databases by building on database design principles such as TFDs. In terms of QA construction, existing benchmarks often rely on manually curated (Zhang & Choi, 2021; Chen et al., 2021; Wei et al., 2023; Kasai et al., 2023; Gupta et al., 2023; Vu et al., 2023; Jia et al., 2024) or template-based QA (Table 9), which often covers limited temporal relations. In contrast, TDBench does not rely on manual construction or a fixed set of templates, while covering a richer set of temporal relations via temporal SQL. There are also LLM-based methods that generate TSQA pairs by processing raw databases via natural language prompting (Kim et al., 2024; Zhao et al., 2024; Chen et al., 2025), but TDBench enables more controlled and precise data processing by employing SQL-based data querying. Lastly, to our knowledge, we are the first to address the automatic evaluation of invalid model explanations in TSQA tasks.

**Using SQL for QA Systems** In QA systems, *Text-to-SQL* has been a primary research area (Qin et al., 2022; Hong et al., 2024), with the goal of (1) translating natural language questions into SQL queries and (2) retrieving answers from databases that serve as knowledge sources, thereby improving QA performance (Pasupat & Liang, 2015; Zhong et al., 2017; Yu et al., 2018; Rajkumar et al., 2022; Liu et al., 2022; Gao et al., 2023). In contrast, our methodology adopts an *SQL-to-Text* approach (Xu et al., 2018; Câmara et al., 2024; Zhang et al., 2024b; Park et al., 2026), which translates SQL queries into natural language questions using translator models such as graph-to-sequence models tailored to the SQL query structure (Xu et al., 2018) or general LLMs (Câmara et al., 2024; Park et al., 2026). Unlike the Text-to-SQL task, which derives SQL queries to improve LLMs' QA performance via SQL-based retrieval, we use the SQL-to-Text task to derive natural language QA pairs for benchmark construction. Furthermore, our key distinction from existing SQL-to-Text approaches is the use of temporal SQL to generate time-sensitive questions tailored to TSQA tasks, leveraging expressive built-in temporal operators to automatically produce diverse temporal contexts.

## 6 CONCLUSION

We proposed TDBench, which utilizes temporal databases and database techniques for systematic evaluation of factual TSQA. By leveraging database techniques, TDBench reduces the manual effort traditionally required in benchmark construction and maintenance, such as designing temporal contexts or updating QA pairs with new world knowledge. We also introduced a new metric called time accuracy for fine-grained TSQA evaluation, designing TDBench to automatically verify both the final answers and the time references by using SQL-based temporal constraints. Experiments revealed that a large portion of factually inconsistent responses with invalid time references can be overlooked in the answer-only evaluation setting, while uncovering model-specific weaknesses across constraint types, answer cardinality, and reasoning hops. We believe TDBench offers a new direction for TSQA evaluation by moving beyond Wikipedia/Wikidata and enabling domain-specific benchmarking.

## 7 LIMITATION

The correctness of the generated QA pairs in TDBench can be affected by the underlying data quality, which is an inherent aspect of data-centric benchmarking approaches (Jia et al., 2018; Jang et al., 2022; Gupta et al., 2023; Kim et al., 2024; Wu et al., 2024). For example, data quality issues like factual inconsistencies and domain-specific anomalies (Wang & Strong, 1996) can lead to noisy QA pairs. While there is extensive work on enhancing data quality (Batini et al., 2009; Chu et al., 2016; Rekatsinas et al., 2017), our study assumes the underlying quality is reasonable and rather focus on an orthogonal goal: enhancing the downstream pipeline to best leverage the available data. Thus, while TDBench provides clear benefits in this context, providing a generalizable and effective tool for converting one's data into a TSQA benchmark, integration with existing data cleaning techniques (Chu et al., 2016; Rekatsinas et al., 2017) would further strengthen the framework. Similarly, we discuss how invalid TFDs affect TDBench in Sec. B.1, while TDBench can exhibit robustness due to its SQL-based grounding.

ETHICS STATEMENT

We believe our research addresses an important aspect of Trustworthy AI by evaluating the accuracy and reliability of LLMs under time-sensitive factual scenarios. In particular, we surface the issue of LLM hallucination in both final answers and time references, which can mislead users and undermine trust. By providing a systematic benchmarking framework to detect such cases, TDBench contributes toward building more reliable and transparent LLMs. As a dynamic framework, the choice of the database in TDBench inherently determines the benchmark content. We assume that it is the user's responsibility to ensure their chosen databases meet ethical standards and exclude harmful or private material. All datasets used in this study are publicly available and free of sensitive or harmful content.

REPRODUCIBILITY STATEMENT

To ensure the reproducibility of our results, we provide the details of experimental setups in the supplementary material, including the proposed TDBench algorithm, system prompts, LLM hyperparameters (e.g., temperature), and details of datasets used in our study. In addition, we will release the GitHub repository for the TDBench framework, containing code and scripts for reproducing our results and documentation for extending the benchmark to new datasets.

THE USE OF LARGE LANGUAGE MODELS

LLMs were mainly utilized as evaluation targets in this work. We also used LLMs to assist in the writing process (e.g., refining phrasing, enhancing clarity), but we did not employ LLMs for generating core research ideas or technical content.

ACKNOWLEDGMENTS

This research was supported by the MSIT(Ministry of Science, ICT), Korea, under the Global Research Support Program in the Digital Field program)(RS-2024-00436680) supervised by the IITP(Institute for Information & Communications Technology Planning & Evaluation). This project is supported by Microsoft Research Asia. This work was also supported by the NYU-KAIST Partnership and by the IITP with a grant funded by the Ministry of Science and ICT (MSIT) of the Republic of Korea in connection with the Global AI Frontier Lab International Collaborative Research. (No. RS-2024-00469482 & RS-2024-00509258). This work was also supported by the National Research Foundation of Korea (NRF) grant funded by the Korea government (MSIT) (No. RS-2022-NR070121).

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

# A APPENDIX – MORE DETAILS ON TEMPORAL DATABASES

## A.1 TEMPORAL DATABASES ACROSS DIVERSE DOMAINS

Continuing from Sec. 2, we introduce examples of temporal databases, widespread across many critical domains:

- *Healthcare*: Electronic Health Records (EHRs), patient timelines, treatment records (Johnson et al., 2016; Pollard et al., 2018)
- *Finance*: Trading records, transaction histories, regulatory data (Zheng et al., 2021; Zhu et al., 2021)
- *Enterprise*: Supply chain, Customer Relationship Management (CRM), Human Resources (HR) databases (Jensen & Snodgrass, 2018)

Public platforms like Kaggle and government data portals also provide readily accessible datasets across diverse domains, as demonstrated in our experiments in Sec. 4 with *Legal, Environmental, Cultural,* and *Social* domains. Despite this prevalence, current TSQA research is highly centered in few data platorms like Wikipedia and Wikidata (Sec. 5), which motivates us to enable the use of these widespread, yet unexplored data for TSQA benchmarking.

## A.2 FORMAL DEFINITIONS OF FDS AND TFDS

Continuing from Sec. 2, we show formal definitions on temporal functional dependencies (TFDs) and temporal joins, the two properties of temporal databases used in our framework.

**Notations**   We begin by introducing the notations used throughout this section. Let $R$ and $S$ be temporal relation schemas, each associated with two timestamps $T_s$ and $T_e$ denoting the start and end of tuple validity, respectively:

$$R = (A_1, \ldots, A_n, C_1, \ldots, C_k, T_s, T_e)$$
$$S = (B_1, \ldots, B_m, C_1, \ldots, C_k, T_s, T_e).$$

For an effective illustration, let $\{A_i\}_{i=1}^n$ and $\{B_i\}_{i=1}^m$ denote the unique attributes of schemas $R$ and $S$, respectively, and let $\{C_i\}_{i=1}^k$ represent the common attributes shared by both relations $R$ and $S$.

**Definition of Temporal functional dependencies (TFDs)**   TFDs are defined following Jensen et al. (1996). Definition A.1 formalizes that, over any time interval $[t_1, t_2]$, tuples that agree on $X$ must also agree on $Y$.

**Definition A.1** (Temporal functional dependencies). *Let $X$ and $Y$ be sets of non-timestamp attributes of a temporal relation schema $R$, and let $t_1$ and $t_2$ be arbitrary times, with $t_1$ not exceeding the current time. A temporal functional dependency, denoted as $X \xrightarrow{T} Y$, exists on $R$ if, for all meaningful instances $r$ of $R$ – that is, database states that are valid, consistent, and free of contradictions – the following holds:*

$$\forall t_1, t_2, \ \forall u_1, u_2 \in \tau_{t_2}(\rho_{t_1}(r)), \ u_1[X] = u_2[X] \implies u_1[Y] = u_2[Y],$$

*where $\rho_{t_1}(r)$ selects tuples valid at $t_1$, and $\tau_{t_2}(\cdot)$ further restricts the result to tuples visible at $t_2$.*

**Temporal Joins**   Temporal natural joins are defined following Jensen & Snodgrass (2018). Intuitively, a temporal natural join combines tuples from two relations that agree on their common attributes and are simultaneously valid in time (i.e., have overlapping validity intervals).

**Definition A.2** (Temporal natural join). *Let $r$ and $s$ be instances of $R$ and $S$, respectively. The temporal natural join of $r$ and $s$, denoted as $r \bowtie^T s$, is defined as:*

$r \bowtie^T s = \{z \mid \exists u \in r, \exists v \in s \, (u[C_1, ..., C_k] = v[C_1, ..., C_k] \land$

$z[A_1, ..., A_n] = u[A_1, ..., A_n] \land z[B_1, ..., B_m] = v[B_1, ..., B_m] \land z[C_1, ..., C_k] = v[C_1, ..., C_k] \land$

$z[T_s, T_e] = overlap(u[T_s, T_e], v[T_s, T_e]) \land z[T_s, T_e] \neq \emptyset)\},$

*where $z$ represents the result tuple, and $overlap(u[T_s, T_e], v[T_s, T_e])$ computes the intersection of the validity intervals from $u$ and $v$.*

---

**Algorithm 1** The proposed `Genqueries` algorithm

---

**Input:** Temporal database relation $s$ with schema $S$, list of temporal functional dependencies (TFDs; Sec. 2) $\mathcal{F} = \{X_1 \xrightarrow{T} Y_1, \ldots, X_n \xrightarrow{T} Y_n\}$ where $X_i$ and $Y_i$ denote sets of attributes in the schema $S$ for $i \in n$, list of temporal relations $\mathcal{R} = \{\text{before}, \text{after}, \ldots, \text{contain}\}$ shown in Table 10
**Output:** List of generated temporal SQL queries $\mathcal{G}$

  1: $\mathcal{G} \leftarrow \emptyset$
  2: **for** each tuple $u \in r$ **do**
  3:     **for** each TFD $f : X_i \xrightarrow{T} Y_i \in \mathcal{F}$ **do**
  4:         *# Phase 1: Generate base query using a TFD*
  5:         $q_b \leftarrow$ SELECT $Y_i$ FROM $s$ WHERE $X_i = u.X_i$
  6:         *# Phase 2: Add temporal constraints*
  7:         **for** each operation $r \in \mathcal{R}$ **do**
  8:             Sample a random time interval $b = (t_s, t_e)$
  9:             $q_t \leftarrow q_b + r(u, b)$ according to Table 10
10:             $\mathcal{G} \leftarrow \mathcal{G} \cup \{q_t\}$
11:         **end for**
12:     **end for**
13: **end for**
14: **return** $\mathcal{G}$

---

In Definition A.2, the first two lines ensure that tuples $u$ and $v$ match on the shared attributes $\{C_i\}_{i=1}^k$ and concatenate the unique attributes $\{A_i\}_{i=1}^n$ and $\{B_i\}_{i=1}^m$, along with a single copy of $\{C_i\}_{i=1}^k$. The third line means the join operation is performed only on tuples with overlapping validity intervals.

# B   APPENDIX – MORE DETAILS ON TDBENCH FRAMEWORK

Continuing from Sec. 3 and Sec. 4, we provide more details on the proposed TDBench framework.

## B.1   MORE DETAILS ON GENQUERIES ALGORITHM

Continuing from Sec. 3.1 and Sec. 3.2, we provide more details on the proposed `Genqueries` algorithm (Alg. 1), which systematically generates temporal SQL queries from an input temporal database. We provide the pseudocode and the potential extension to a more diverse temporal scenarios.

**Algorithm Procedure of `Genqueries`**   We explain the `Genqueries` algorithm (Alg. 1) line by line. `Genqueries` begins by initializing an empty list of queries (Line 1). For each tuple in the database and each TFD defined in the database (Lines 2–3), we construct a base SQL query by selecting the $Y_i$ attributes conditioned on the $X_i$ attributes of the tuple (Line 4). We then add temporal conditions to this base query (Line 5) by iterating over a set of predefined temporal operations, such as "before", "after", or "overlap" (Line 6), using the relation-SQL condition mapping in Table 10. For each operation, we sample a random time interval (Line 7), adding the temporal constraint to the base query (Line 8), and append the resulting query to the output list (Line 9). After processing all tuples and TFDs, `Genqueries` returns the complete set of generated temporal SQL queries (Line 10).

**Extending `Genqueries`**   We can further extend `Genqueries` to incorporate more diverse data attributes, data types, and TSQA evaluation scenarios.

- *Attributes beyond TFD-selected attributes*: `Genqueries` also supports user-defined extensions where users can manually specify alternative sets of question and answer attributes. For example, given the relation *Leader(country, role, name, gender, start, end)* (Table 1) and the TFD $country, role \xrightarrow{T} name$, one can extend the answer to include both $name$ and $gender$, while still using $country, role$ as the question. Although this extension requires manual input to specify attribute sets, `Genqueries` still greatly simplifies the QA construction process compared to manual construction or template creation.

- *Semi-strutured data*: Although not our primary focus, `Genqueries` can be readily applied to some semi-structured data by transforming this data into structured formats. For example, techniques like schema inference and normalization (DiScala & Abadi, 2016; Baazizi et al., 2019) can be utilized during the data preprocessing stage.

- *Other temporal QA tasks*: In addition, SQL's expressiveness allows flexible extensions to richer question types. For example, aggregation operators (e.g., GROUP BY, HAVING) enable event-counting questions like "Which president has been elected for the third time?", which can be generated from the condition "GROUP BY... HAVING COUNT(*)=3" and require the model to count specific events throughout history.

## B.2   GENERATED SQL CONDITIONS AND TIME ACCURACY CRITERIA

Continuing from Sec. 3 and Sec. 4, we present the complete set of SQL conditions used in TDBench, derived from Allen's interval algebra (Allen, 1983). Table 10 summarizes each temporal relation, its corresponding SQL implementation, an example temporal context, and the criteria used for time accuracy evaluation (Sec. 3.2).

Table 10: **The complete set of SQL conditions for temporal context generation using Allen's interval algebra.** Each interval relation is implemented as an SQL condition to create temporal contexts. We use $a$ as the validity interval of a tuple (such as a president's term), while $b$ is randomly sampled to generate various contexts according to the interval relation. The overlap relation has two implementations: one for the temporal reasoning task and the another for the "current" condition in the temporal alignment task (see more details in Sec. B.2). We also show relation-specific evaluation criteria for evaluating time accuracy.

| Relation | Interval Diagram | SQL Condition | Example Temporal Context | Criteria |
|---|---|---|---|---|
| $a$ before $b$ | | $a$.end < $b$.start | A president who ends before *September 20, 2000* | end |
| $a$ after $b$ | | $a$.start > $b$.end | A president who starts after *March 20, 2001* | start |
| $a$ meet $b$ | | $a$.end = $b$.end - $b$.length | A president who ends exactly *half a year* before *March 20, 2001* | end |
| $a$ met-by $b$ | | $a$.start = $b$.start + $b$.length | A president who starts exactly *half a year* after *September 20, 2000* | start |
| $a$ overlap $b$ | | $a$.start < $b$.start $\land$ b.start < a.end < b.end | A president who starts before *September 20, 2000* and ends between *September 20, 2000* and *March 20, 2001* | start, end |
| | | $a$.start < $b$.start $\land$ $a$.end IS NULL | A president who is currently serving | start |
| $a$ overlapped-by $b$ | | $a$.end > $b$.end $\land$ b.start < a.start < b.end | A president who starts between *September 20, 2000* and *March 20, 2001* and ends after *March 20, 2001* | start, end |
| $a$ equal $b$ | | $a$.start = $b$.start $\land$ $a$.end = $b$.end | A president who starts in *September, 2000* and ends in *March, 2001* | start, end |
| $a$ start $b$ | | $a$.start = $b$.start $\land$ $a$.end < $b$.end | A president who starts in *September, 2000* and ends before *March, 2001* | start, end |
| $a$ started-by $b$ | | $a$.start = $b$.start $\land$ $a$.end > $b$.end | A president who starts in *September, 2000* | start |
| $a$ finish $b$ | | $a$.start > $b$.start $\land$ $a$.end = $b$.end | A president who starts after *September 20, 2000* and ends in *March, 2001* | start, end |
| $a$ finished-by $b$ | | $a$.start < $b$.start $\land$ $a$.end < $b$.end | A president who ends in *March, 2001* | end |
| $a$ during $b$ | | $a$.start > $b$.start $\land$ $a$.end < $b$.end | A president who starts after *September 20, 2000* and ends before *March 20, 2001* | start, end |
| $a$ contain $b$ | | $a$.start < $b$.start $\land$ $a$.end > $b$.end | A president who starts before *September 20, 2000* and ends after *March 20, 2001* | start, end |

**Handling Time Constraint of "Current"**  In Table 10, we note that the *overlap* relation is modeled with two separate SQL implementations to support both temporal alignment and temporal reasoning task scenarios. For temporal alignment, which evaluates whether LLMs provide up-to-date knowledge, it is necessary to represent the "current" time condition. Depending on the data format, the "current" condition can be encoded by setting the end date to a special value, such as a maximum date 99-99-99 or NULL. We adopt the NULL format because it produces more readable SQL predicates (e.g., `end IS NULL`) and simplifies the next step of SQL-to-text conversion process (Fig. 1). With this approach, the "current" condition is treated as a special case of the overlap relation, where the interval starts before the present and has no defined end. This dual implementation allows us to flexibly generate both current (temporal alignment) and general (temporal reasoning) temporal contexts.

### B.3  HANDLING MISSING, UNKNOWN, OR INVALID TFDS

Continuing from Sec. 3, we discuss how to handle cases where TFDs are missing, unknown, or invalid in the input temporal database of TDBench framework.

**Missing TFDs**  If TFDs are not present in the database schema, users can manually specify which attribute sets to use for question and answer generation. While this approach allows TDBench to be flexibly applied even without explicit TFDs, it introduces additional manual cost.

**Unknown TFDs**  When TFDs are not known in advance, users can employ automated FD/TFD discovery tools (e.g., Dep-Miner (Lopes et al., 2000), FastFDs (Agrawal et al., 1993), DFD (Abedjan et al., 2014), and the Python library FDTool (Buranosky et al., 2019)) to infer candidate dependencies from the data. This is particularly useful for large or complex schemas, or when users are unfamiliar with the database structure.

**Invalid TFDs**  If a TFD does not strictly hold due to data anomalies or noise, the generated QA pairs still reflect the actual database contents. This is because, as illustrated in Table 3, TDBench utilizes SQL-based grounding when constructing QA pairs, so all answers are consistent with the underlying database. However, in such cases, the answer set may include multiple valid answers, which may compromise the original intent of unique-answer questions. In more complex multi-hop settings, the lack of valid TFDs can lead to incorrect QA construction. To prevent such cases, users are encouraged to clean the data and validate TFDs in advance. Our official implementation provides Python code for automatic TFD validation to help users detect and address these issues.

### B.4  MORE DETAILS ON NATURAL LANGUAGE QA CONVERSION

Continuing from Sec. 3, we provide more details on the natural language conversion step, where we use GPT-4o (OpenAI, 2024) as an SQL-to-text translator.

**System Prompt for SQL-to-text**  The following prompt is an example system prompt used in our zero-shot setting for SQL-to-text conversion. We set the temperature parameter of GPT-4o to 0.3 to promote diverse paraphrasing. The prompt instructs the model to convert SQL queries into natural language questions based on the given schema and generate three different phrasings per query, where more than three questions can be generated if desired by modifying the prompt accordingly.

```
The following SQL query is about a relational database Leader(country,
role, name, start, end), where start and end represent date information.
Convert the provided SQL query into a natural language question.
Generate three different questions, each starting with Q:.
Only return the generated questions.
```

**Accuracy of SQL-to-text**  We assess the accuracy of SQL-to-text conversion in the above setup by randomly sampling 130 generated questions during the conversion and manually verifying whether the questions correspond to the underlying query. We observe 91.5% accuracy, consistent with recent findings that modern LLMs have strong zero-shot performance on SQL-to-text tasks (Zhang et al., 2024a). We analyze error cases as follows:

- *Prime ministers of Japan → Japanese prime minister*: Although a prime minister of a country may not necessarily be a citizen, the model often assumes nationality based on the position, making this the most common error type.
- *November 1st → November*: In some cases, specific dates (e.g., November 1st) are generalized to the corresponding month (e.g., November), introducing potential ambiguity.
- *Calculation error*: When processing temporal operations like DATE('2006-07-17 00:00:00', '-2 months') in the query, the model occasionally makes date calculation errors. However, such errors are relatively rare, occurring in fewer than 1% of cases.

**Examples of Generated QA pairs**    In addition to the example QA pairs shown in the main text (Table 3) generated from the "meet" relation, we present additional examples across temporal relations. Table 11 and Table 12 show QA pairs derived from the "contains" and "overlap" relations, respectively, which are two examples of temporal relations rarely covered in prior TSQA benchmarks. Table 13 shows QA pairs from an additional SQL condition for the "overlap" relation, specifically designed to target current data (see Sec. B.2 for more details). Table 14 shows multi-hop QA pairs generated by using temporal joins (Sec. A.2).

Table 11: **QA pair generation from an SQL query (*'contain' relation*)**. The query is translated to multiple natural language questions with the same answer. To evaluate both answer and time accuracy, the query also outputs a relation-specific time reference (e.g., start and end for the *'contain'* relation), defined in Table 10.

| SQL Query (relation="contain") | Generated QA |
|---|---|
| `SELECT name, start, end FROM Leader WHERE Country='Japan' AND Role='Emperor' AND Start < '2019-10-01' AND End > '2024-08-26'` | **[Questions]** "Who was the Emperor of Japan whose term started before October 1, 2019, and ended after August 26, 2024?", "What is the name of the Emperor of Japan who began their role before October 1, 2019, and ended it after August 26, 2024?"... 
 **[Answer]** Akihito 
 **[Time reference]** 1989-01-07 (start), 2019-05-01 (end) |
| **[Model Response (Gemma2-27B)]** *Akihito*. He reigned from *1989-01-07* to *2019-05-01*. | |

Table 12: **QA pair generation from an SQL query (*'overlap' relation that models basic temporal reasoning*)**. The query is translated to multiple natural language questions with the same answer. To evaluate both answer and time accuracy, the query also outputs a relation-specific time reference (e.g., start and end for the *'overlap'* relation), defined in Table 10.

| SQL Query (relation="overlap") | Generated QA |
|---|---|
| `SELECT name, start, end FROM Leader WHERE Country='Germany' AND Role='President' AND Start < '2003-09-30' AND End BETWEEN '2003-09-30' and '2007-06-30'` | **[Questions]** "Who was the president of Germany who started their term before September 30, 2003, and ended their term between September 30, 2003, and June 30, 2007?", "Can you list the names of the presidents of Germany whose terms began before September 30, 2003, and ended between September 30, 2003, and June 30, 2007?"... 
 **[Answer]** Johannes Rau 
 **[Time reference]** 1999-07-01 (start), 2004-06-30 (end) |
| **[Model Response (Qwen2-72B)]** *Johannes Rau*, who started his term on *July 1, 1999*, and ended it on *June 30, 2004*. | |

### B.5   FORMAL DEFINITION OF TIME ACCURACY

Continuing from Sec. 3.2, we present the formal definition of the proposed time accuracy metric.

We begin by defining the *set of time references* $\mathcal{T}$, which captures the time references to be verified in LLM responses. Since we primarily generate questions asking about real world events – each grounded with a start date and end date – we set $\mathcal{T} = \{t_s, t_e\}$, where $t_s$ and $t_e$ denote the start and end dates, respectively.

Table 13: **QA pair generation from an SQL query (*'overlap'* relation that models the temporal constraint of "current")**. The query is translated to multiple natural language questions with the same answer. To evaluate both answer and time accuracy, the query also outputs a relation-specific time reference (e.g., `start`), defined in Table 10.

| SQL Query (relation="overlap") | Generated QA |
|---|---|
| `SELECT name, start`
`FROM Leader WHERE`
`Country='Netherlands' AND`
`Role='King' AND End IS NULL` | **[Questions]** "Who is currently serving as the King of the Netherlands?", "What is the name of the individual who is the reigning King in the Netherlands?"...

**[Answer]** Willem-Alexander
**[Time reference]** 2013-04-30 (start) |
| **[Model Response (Llama3.1-70B)]** King *Willem-Alexander*, since *April 2013*. | |

Table 14: **QA pair generation from an SQL query (for a joined table).** The query is translated to multiple natural language questions with the same answer. To evaluate both answer and time accuracy, the query also outputs timestamp attributes (e.g., `start`, `end`). $\bowtie^T$ denote the temporal join operator, which only joins tuples with overlapping validity intervals – see formal definition of temporal join in Sec. A.2.

| SQL Query (temporal join) | Generated QA |
|---|---|
| `SELECT name, start, end FROM`
`Leader` $\bowtie^T$ `Olympic WHERE`
`game_edition='7th Winter'`
`AND Role='President'` | **[Questions]** "Who was the president of the host country for the 7th Winter Olympic Games?", "Can you provide the name of the president of the host country during the 7th edition of the Winter Olympics?"...
**[Answer]** Giovanni Gronchi
**[Time reference]** 1955-05-11 (start), 1962-05-11 (end) |
| **[Model Response (GPT-4o)]** The 7th Winter Olympic Games were held in Cortina d'Ampezzo, Italy in February 1956. The President of Italy at that time was *Giovanni Gronchi*, who served from *May 1955* to *May 1962*. | |

We then define the *question-reference function* $f : q \rightarrow \mathcal{T}$, which maps a question to the time references to be verified. This function can be flexibly designed based on the temporal contexts expressed in the question. In our setup, the underlying SQL queries explicitly specify which time reference are relevant to answer selection process (e.g., an SQL condition such as "start < January 2001" implies reliance on start date $t_s$). Accordingly, we define $f$ with respect to the temporal relation used in the SQL query, as summarized in Table 10.

We finally define the time accuracy **T** given a question $q$ as follows:

$$\mathbf{T}(q) = \frac{|\{t \in f(q) \mid t \in \text{model response}\}|}{|f(q)|} \times 100(\%). \tag{1}$$

Note that for temporal relation types requiring multiple time references (e.g., *'contain'* in Table 11), partial credit is given when only a subset is correctly included in the model response.

### B.6 TIME ACCURACY EVALUATION WITH LLM-JUDGE

Continuing from Sec. 3.2, we provide more details on time accuracy evaluation, where we use Deepseek-R1-14B (Guo et al., 2025) as an LLM-judge.

**System Prompt for LLM-Judge**  The following system prompt shows the instruction for time accuracy evaluation, consistent with the formal definition in Eq. 1. As shown in the prompt, we opt to use LLM-Judge to capture various expressions of time references (e.g., "26 Jan 2025", "2025/01/26"), which may be missed by exact-match methods that directly compares gold time reference strings ({entity_date}) against the evaluated model response ({response}).

```
You are given a reference **start date** and **end date**.
Check whether the response correctly includes both dates, even if they
are expressed in a different but equivalent format (e.g., '26 Jan 2025',
'January 26, 2025', '2025/01/26', etc.).

- If **both** of the two dates is are correctly mentioned with the
intended meaning (i.e., the start date is described as the start date,
and the end date as the end date), respond with **"Yes"**.
- If **one** of the two dates is correctly mentioned with the intended
meaning, respond with **"Half"**.
- If **neither** date is correctly mentioned with the correct meaning,
respond with **"No"**.
Your answer must be one of: 'Yes', 'Half', or 'No'. Be concise.

{entity_date}

**Response:**
{response}

**Answer:
```

**Accuracy of LLM-Judge in Time Accuracy Evaluation**   Similarly to the manual verification setup in Sec. B.4, we assess the effectiveness of LLM-Judge in time accuracy evaluation by randomly sampling 130 responses and manually verifying whether the evaluation aligns with the instruction above. We observe 91.1% accuracy, with most errors arising from instruction misinterpretation (e.g., assessing end dates when only assessing start dates is required) and date match failures (e.g., failing to match the gold time reference of "1993-04-28" and "April 1993" in the model response).

We note that using stronger LLMs can further enhance evaluation correctness; for instance, we observe that using GPT-4o under the same setup yields higher accuracy (95.5%). However, we opt not to use GPT-4o in our main experiments, as it is one of the evaluated models and may incur potential bias during evaluation if used as a judge.

**Extending Time Accuracy to Non-standard Temporal Units**   While TDBench does not consider non-standard temporal units (e.g., partial months, fiscal quarters) as first-class representations, TDBench is highly extensible as follows:

- *Open intervals* using null formats (e.g., [2025-11-14, NULL]), as utilized in generating temporal alignment questions (Sec. B.2).
- *Granularities beyond date* using a mapping to a set of time ranges. For example, mapping FY2020 Q1 $\Rightarrow$ [2020-01-01, 2020-03-31], FY2020 Q2 $\Rightarrow$ [2020-04-01, 2020-06-30] for fiscal quarters, and "early June 2020" $\Rightarrow$ [2020-06-01, 2020-06-10] for partial months.
- *Finer-grained time representation* using built-in database data types (e.g., `2020-01-01 13:10:01` via `TIMESTAMP` type).

In addition, updating the benchmark to incorporate such granularities is easy for TDBench, where only database-level preprocessing is required, and TDBench will automatically regenerate the benchmark. Extending to a more nuanced temporal information such as uncertain intervals remains as a valuable future direction.

## B.7   ADDITIONAL CONTEXT CONSTRUCTION

Continuing from Sec. 3.2, we provide more details on context construction used in the open-book setting. As illustrated in Fig. 4, we append both relevant rows and irrelevant rows from the table with the gold answer to effectively evaluate LLMs' temporal reasoning abilities given a time-sensitive question. Relevant rows are retrieved by (1) removing the temporal condition from the underlying SQL query of the question and (2) executing the modified query on the database, returning entities across different periods (e.g., all U.S. presidents when the question asks about the U.S. president in 2019). We randomly sample the irrelevant rows in the table.

```
| Country | Role | Name | Start | End |                    Open-book QA
| --------- | ----- | -------- | ----- | --- |
| India | President | Sharma | 1992-07-25 | 1997-07-25 |    ⎫
| India | President | Kovind | 2017-07-25 | 2022-07-25 |    ⎬ Relevant
| India | President | Narayanan | 1997-07-25 | 2002-07-25 |  ⎭ rows
| France | President | Chirac | 1995-05-17 | 2007-05-16 |   ⎫
| Egypt | Prime Minister | Madbouly | 2018-06-14 | nan |    ⎬ Irrelevant
| Austria | Chancellor | Bierlein | 2019-06-03 | 2020-01-07 | ⎭ rows

Q: Who was the President of India that started his term after May 31,
1996, and ended his term between July 1, 2002, and July 31, 2002?

A:                                                       Closed-book QA
```

Figure 4: **Open-book vs. Closed-book QA in TDBench.** The open-book setting provides table rows as additional context, while the closed-book setting provides only the question.

## C  APPENDIX – MORE DETAILS ON EXPERIMENTAL SETUPS

Continuing from Sec. 4, we provide more details on the experimental settings and datasets used.

### C.1  EXPERIMENTAL SETUPS AND SYSTEM PROMPTS

**Experiment Settings**  We use the following LLM versions: GPT-3.5 (2024-05-01-preview), GPT-4 (2025-01-01-preview), GPT-4o (2024-02-01), Llama3.1-70B (2024-07-23), Mixtral-8x7B (2023-12-11), Gemma2-27B (2024-06-27), Qwen2-72B (2023-08-03), and Granite3.1-8B (2024-12-18). All experiments are conducted on NVIDIA Quadro RTX 8000 GPUs.

**System Prompts for TSQA Tasks**  We use the following system prompts for the TSQA tasks: the upper prompt is used for the temporal alignment task (Sec. 4.1), and the lower prompt is used for the temporal reasoning task (Sec. 4.2). As the prompts explicitly require time information in the rationale, responses that omit such references are marked incorrect in the time accuracy evaluation. This policy is applied uniformly across all models to ensure consistent and fair TSQA benchmarking, reflecting our goal of evaluating both final answers and explanations.

```
Answer the following question. Provide the short direct answer that
includes both the factual answer and the date information (month and
year) of the fact, prefixed with the word 'since'. Say 'unsure' if you
don't know. Be concise.
```

```
Answer the following question. Provide the short direct answer that
includes both the factual answer and rationale with the date information
of the fact. If there is no valid answer, respond with 'No answer'.
If there is one correct answer, return it as a short sentence.
If there are multiple valid answers, present them clearly as bullet
points. If you are unsure, respond with 'unsure'. Be concise.
```

### C.2  MORE DETAILS ON DATASETS

Continuing from Sec. 4, we provide more details on the temporal databases used in our experiments. Table 15 summarizes the schema, selected TFDs, and the number of questions generated from each dataset. Table 16 presents example questions generated by converting TFDs. Using these datasets, we construct 2,079 temporal alignment, 6,756 temporal reasoning, and 396 multi-hop questions, with additional questions readily generable if needed.

**Wikipedia**  We select domains following the conventional TSQA benchmarking literature (Dhingra et al., 2022; Margatina et al., 2023; Tan et al., 2023; Mousavi et al., 2024), including *Country*, *Athlete*, *Organization*, and *Olympic* datasets. We also follow the entity selection step of Mousavi et al. (2024), which select entities likely to appear in most LLM training corpora to reduce performance variance caused by entity frequency when retrieving facts (Mallen et al., 2023; Sun et al., 2023). All datasets were retrieved on 02 January 2025.

Table 15: **Details of the temporal databases used in our experiments (Sec. 4).**

| Data Source | Question # | Schema | TFD |
|---|---|---|---|
| Wikipedia | 6,177 | *Country(country, role, name, start, end)* | *country, role* $\xrightarrow{T}$ *name* |
| | | *Athlete(name, team, sport, start, end)* | *name, sport* $\xrightarrow{T}$ *team* |
| | | *Organization(org_name, type, role, name, start, end)* | *org_name, type, role* $\xrightarrow{T}$ *name* |
| | | *Olympic(game_round, game_edition, city country, season, game_name, start, end)* | *game_round, role* $\xrightarrow{T}$ *name (joined)* 
 *game_edition, role* $\xrightarrow{T}$ *name (joined)* |
| Domain-specific | 1,704 | *Law(country, law_type, legality, start, end)* | *country, law_type* $\xrightarrow{T}$ *legality* |
| | | *Carbon Tax(jurisdiction, tax_type, instrument_name, status, start, end)* | *jurisdiction, type* $\xrightarrow{T}$ *status* |
| | | *Heritage(heritage_element, member_state, status, region, start, end)* | *heritage_element* $\xrightarrow{T}$ *status* |
| | | *Netflix(title, director, cast, release_year, popularity, start, end)* | *title* $\xrightarrow{T}$ *director* |
| Synthetic | 1,350 | *Medical(name, gender, blood_type medication, doctor, hospital, insurance_provider, billing_amount)* | *name, gender, blood_type* $\xrightarrow{T}$ *doctor* 
 *name, gender, blood_type* $\xrightarrow{T}$ *medication* |

Table 16: **Examples of base questions generated from TFDs across datasets.** The proposed `Genqueries` algorithm (Alg. 1) converts the TFDs listed in Table 15 into these base questions and augments them with diverse temporal constraints.

| Data Source | Example Questions Generated From TFDs |
|---|---|
| Wikipedia | • *Country*: Who is serving as the president of Italy? 
 • *Athlete*: What team did Stephen Curry play for in basketball? 
 • *Organization*: Which person began their tenure as general secretary at the United Nations organization? 
 • *Olympic*: Can you provide the name of the president of the host country during the 7th edition of the Winter Olympics? |
| Domain-specific | • *Law*: Is joint adoption by same-sex couples legal or illegal in Argentina? 
 • *Carbon Tax*: Is the carbon tax mechanism in Finland active? 
 • *Heritage*: Is the heritage element "Cheoyongmu" inscribed or proclaimed? 
 • *Netflix*: Who directed the most recent release of the movie titled "Attack on Titan"? |
| Synthetic | • Who was the doctor for Jerry Martin, a male patient with blood type A+? 
 • Which medication is given to Kenneth Jacobs, a female patient with AB+ blood type? |

**Domain-specific** To demonstrate the applicability of TDBench beyond Wikipedia-based datasets, we model domain-specific TSQA scenarios across four contexts: *Legal*, *Environmental*, *Cultural*, and *Social*. These domains are represented by datasets on same-sex marriage laws sourced from the ILGA World databases[2], a global knowledge base on legal frameworks and human right bodies, retrieved on 02 January 2025; carbon taxation data sourced from the World Bank Group's "State and Trends of Carbon Pricing Dashboard"[3], retrieved on 17 October 2024; UNESCO heritage elements sourced from the official UNESCO website[4], retrieved on 17 October 2024; and Netflix TV shows data sourced from Kaggle[5], retrieved on 05 May 2025. For the Netflix dataset, we only consider TV series that share the same title, but have been remade over time with different directors.

---

[2] https://database.ilga.org/en

[3] https://carbonpricingdashboard.worldbank.org/compliance/instrument-detail

[4] https://whc.unesco.org/en/list/

[5] https://www.kaggle.com/code/sonawanelalitsunil/netflix-movies-tv-shows-till-2025

**Synthetic** We also show how TDBench can be applied to synthetic datasets by using the *Medical* dataset, which contains synthetic patient records sourced from Kaggle[6], retrieved on 05 May 2025.

# D  APPENDIX – MORE EXPERIMENTAL RESULTS

Continuing from Sec. 4, we provide more details on experimental results.

## D.1  ERROR ANALYSIS ON TEMPORAL ALIGNMENT EVALUATION

Continuing from Sec. 4.1, we present example error cases in the temporal alignment task, where LLMs often hallucinate time references while providing correct answers. Table 17 and Table 18 show such responses from the Heritage and Netflix datasets, respectively, where LLMs generate inaccurate time references (e.g., "December 2011" and "January 2020") alongside correct answers (e.g., "Inscribed" and "Mehmet Ada Öztekin"). These cases indicates that LLMs struggle to accurately align the time references with the factual answers, highlighting that capturing temporal misalignment in model explanations can be a critical yet overlooked issue in the current temporal alignment task.

Table 17: **Example LLM responses on the temporal alignment task using the Heritage dataset.** The time granularity used for time accuracy is year.

| |
|---|
| **[Question]** Is the UNESCO heritage element "Cheoyongmu" currently inscribed or proclaimed? 
 **[Answer]** Inscribed 
 **[Time reference]** November 2009 (start) |
| **[Model Response]** 
 GPT-3.5: Since Inscribed (✓) December 2011 (✗). 
 GPT-4: Inscribed (✓) since November 2009 (✓). 
 GPT-4o: Inscribed (✓) since November 2009 (✓). 
 Llama3.1-70B: Inscribed (✓) since November 2009 (✓). 
 Mixtral-8x7B: Proclaimed (✗) since December 2017 (✗). 
 Gemma2-27B: Inscribed (✓) since November 2022 (✗). 
 Qwen2-72B: "Cheoyongmu" has been inscribed (✓) on UNESCO's list since 2015 (✗). 
 Granite3.1-8B: Inscribed (✓) since 2019 (✗) |

Table 18: **Example LLM responses on the temporal alignment task using the Netflix dataset.** The time granularity used for time accuracy is year.

| |
|---|
| **[Question]** Which director was involved in the most recent version of "Miracle in Cell No. 7"? 
 **[Answer]** Mehmet Ada Öztekin 
 **[Time reference]** October 2019 (start) |
| **[Model Response]** 
 GPT-3.5: Since January 2020 (✗), the director of the latest release of "Miracle in Cell No. 7" is Mehmet Ada Öztekin (✓). 
 GPT-4: Since March 2020 (✗), the director of the most recently released movie "Miracle in Cell No. 7" is Mehmet Ada Öztekin (✓). 
 GPT-4o: Since October 2023 (✗), the director of the most recent "Miracle in Cell No. 7" movie is Mehmet Ada Öztekin (✓). 
 Llama3.1-70B: Lee Hwan-kyung (✗), since January 2019 (✓). 
 Mixtral-8x7B: Sure, since January 2019 (✓), the director of the latest release of "Miracle in Cell No. 7" is Mehmet Ada Öztekin (✓). 
 Gemma2-27B: Since August 2023 (✗), the director of the latest release of "Miracle in Cell No. 7" is Lee Hwan-kyung (✗). 
 Qwen2-72B: The original film was released in 2013 and a Chinese remake directed by Zhang Lü (✗) was released in January 2021 (✗). 
 Granite3.1-8B: The most recent version of ""Miracle in Cell No. 7"" was directed by Lee Jae-gon (✗), since 2019 (✓). |

---

[6]https://www.kaggle.com/datasets/prasad22/healthcare-dataset

## D.2  Integrating Time Accuracy to Existing TSQA Benchmarks

Continuing from Sec. 4.1, we provide a detailed case study with Dyknow (Mousavi et al., 2024), one of the TSQA benchmarks that employs open-ended QA templates (e.g., "The president of Italy is __") (Dhingra et al., 2022; Margatina et al., 2023) to assess whether LLMs are temporally aligned with the current world.

While Dyknow evaluates LLM responses as correct (Cor.), outdated (Out.), or incorrect (Inc.) based on the parsed answer, we modify their system prompt to explicitly generate start date references to incorporate time accuracy. We then assess the correctness of each evaluation method (i.e., with and without time accuracy) against manual verification on 125 randomly sampled responses per model, where we recruit five graduate student annotators and distribute a total 1,000 responses from eight LLMs, ensuring that each response is evaluated by at least two annotators. We measure the correctness with precision (P), recall (R), and F1-score (F1), standard metrics for classification (Yacouby & Axman, 2020) with definitions provided below.

- Precision: The proportion of examples labeled as a given class (e.g., correct, outdated, and incorrect) by the evaluation method that are also judged as the same class by human annotators.

$$\text{Precision} = \frac{\text{True Positives}}{\text{True Positives} + \text{False Positives}}$$

- Recall: The proportion of examples judged to a given class (e.g., correct, outdated, and incorrect) by human annotators that are also labeled as the same class by the evaluation method.

$$\text{Recall} = \frac{\text{True Positives}}{\text{True Positives} + \text{False Negatives}}$$

- F1 Score: The harmonic mean of precision and recall, providing a balanced measure of alignment with human judgment.

$$\text{F1 Score} = 2 \times \frac{\text{Precision} \times \text{Recall}}{\text{Precision} + \text{Recall}}$$

As shown in Table 19, integrating $\mathbf{T}$ consistently improves all metrics over the original method, leading to higher evaluation correctness. These results demonstrate the value of time accuracy as a complementary evaluation criterion to traditional answer accuracy. We note that the integration is straightforward for other TSQA benchmarks that employ open-ended question formats similar to Dyknow (Dhingra et al., 2022; Margatina et al., 2023), requiring only minimal prompt modifications.

Table 19: **QA Evaluation correctness of DyKnow (Mousavi et al., 2024), with and without time accuracy metric (T).** Human evaluation correctness is used as the ground truth.

|  | Methods | P | R | F1 |
|---|---|---|---|---|
| **Correct** ($\uparrow$) | Human | 1.00 | 1.00 | 1.00 |
|  | Dyknow | 0.67 | 0.92 | 0.77 |
|  | Dyknow + **T** | **0.98** | **0.95** | **0.96** |
| **Outdated** ($\downarrow$) | Human | 1.00 | 1.00 | 1.00 |
|  | Dyknow | 0.69 | 0.78 | 0.73 |
|  | Dyknow + **T** | **0.95** | **0.88** | **0.91** |
| **Incorrect** ($\downarrow$) | Human | 1.00 | 1.00 | 1.00 |
|  | Dyknow | 0.32 | 0.60 | 0.39 |
|  | Dyknow + **T** | **0.58** | **0.84** | **0.64** |

## D.3  Temporal Span Analysis Aggregated Across Multiple Domain

Continuing from Sec. 4.2, we provide more results on temporal span analysis. In addition to Fig. 3 that shows LLM performances by temporal span evaluated on a single domain, we show aggregated results in Fig. 5 evaluated on multiple domains: Country, Athletes, and Organizations. Compared to the single domain results, LLM performances diverge more clearly, suggesting that domain-specific fluctuations are smoothed out and model-specific behaviors become more evident when multiple

domains are considered. General trends – notable performance drops in recent years (2020-2025) while having performance peaks in 2010-2015 – are still observed, which are consistent with recent findings (Dhingra et al., 2022; Mousavi et al., 2024; Zhao et al., 2024).

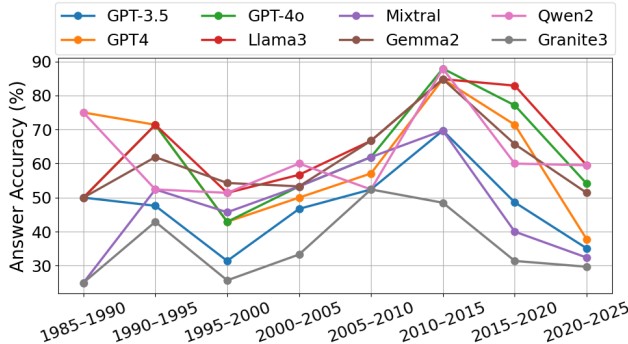

Figure 5: **LLM performances across different time spans (1985-2025) in the closed-book setting, evaluated on multiple domains:** *Country, Athletes*, **and** *Organizations*.

### D.4 MORE DETAILS ON EXPERIMENTS WITH PERFORMANCE IMPROVEMENT TECHNIQUES

Continuing from Sec. 4.3, we provide more details on experiments with performance improvement techniques: RAG (Lewis et al., 2020), CoT (Wei et al., 2022), and Time-CoT (Yin & Hu, 2024).

**CoT and Time-CoT Prompts** The upper and lower prompts are used for the CoT and Time-CoT settings, respectively. As illustrated, Time-CoT prompts provide more detailed intermediate steps for temporal reasoning, such as presenting relevant historical timelines (e.g., sequences of presidential terms) when compared to standard CoT.

```
Q: Who was the vice president of the host country for the 31th Summer
Olympic Games?
A: Michel Temer. The 31th Summer Olympic Games were held in August 2016
in Rio. The host country was Brazil. The Vice President of Brazil in
August 2016 was Michel Temer. The answer is Michel Temer.

Q: Can you provide the name of the governor-general of the host country
during the 27th edition of the Summer Olympics?
A: William Deane. The 27th edition of the Summer Olympics were held
in September 2000 in Sydney. The host country was Australia.
The Governor-General of Australia in September 2000 was William Deane.
The answer is William Deane.

Q: Can you identify the president of the host country for the Seoul 1988
Olympics?
A: Roh Tae-woo. The Seoul 1988 Olympics were held in September 1988.
The host country was South Korea. The president of South Korea in
September 1988 was Roh Tae-woo. The answer is Roh Tae-woo.

Q: Who held the position of general secretary in the host country during
the 1980 Olympic Games in Moscow?
A: Leonid Brezhnev. The Moscow 1980 Olympics were held in July 1980.
The host country was Soviet Union. The general secretary of Soviet Union
in July 1980 was Leonid Brezhnev. The answer is Leonid Brezhnev.
```

```
Q: Can you identify the president of the host country for the Seoul
1988 Olympics?
A: Let's think step by step. First, Seoul 1988 Olympic Games were held
in September 1988, South Korea. Then, we need to find out who was
the president of South Korea according to timeline. 1989-1993: Roh
```

```
Tae-woo, 1993-1998:Kim Young-sam. Then, we need to focus on the time
-- 1988. So we can obtain the answer is Roh Tae-woo.
Therefore, the answer is Roh Tae-woo.

Q: Who held the position of general secretary in the host country during
the 1980 Olympic Games in Moscow?
A: Let's think step by step. First, Moscow 1980 Olympic Games were
held in July 1980, Soviet Union. Then, we need to find out who was
the general secretary of Soviet Union according to timeline. 1964-1982:
Leonid Brezhnev, 1982-1984: Yuri Andropov. Then, we need to focus on
the time -- 1980. So we can obtain the answer is Leonid Brezhnev.
Therefore, the answer is Leonid Brezhnev.

Q: What is the name of the monarch of the host country for the Oslo 1952
Olympic Games?
A: Let's think step by step. First, Oslo 1952 Olympic Games were held
in February 1952, Norway. Then, we need to find out who was the monarch
of Norway according to timeline. 1905-1957: Haakon VII, 1957-1991: Olav
V. Then, we need to focus on the time -- 1952. So we can obtain the
answer is Haakon VII. Therefore, the answer is Haakon VII.

Q: Which emperor was in charge of the host country during the Sapporo
1972 Olympic Games?
A: Let's think step by step. First, Sapporo 1972 Olympic Games were
held in February 1972, Japan. Then, we need to find out who was the
president of Japan according to timeline. 1926-1989: Showa, 1989-2019:
Akihito. Then, we need to focus on the time -- 1972. So we can obtain
the answer is Showa. Therefore, the answer is Showa.
```

**Case Study of Temporally Misaligned RAG** We observe that standard RAG methods often fail to account for temporal cues embedded in the question. Table 20 and Table 21 present illustrative examples of such temporal misalignments, where retrieved documents from Wikipedia describe the Olympics from incorrect time periods or provide generic explanations of the Olympics.

Table 20: **Retrieved Wikipedia passage describing the 1964 Winter Olympics, given the question asking about the 7th Winter Olympics (held in 1956).**

| |
|---|
| **[Question]** Who was the president of the host country for the 7th Winter Olympic Games? |
| **[Retrieved Document]** The 1964 Winter Olympics, officially known as the IX Olympic Winter Games (German: IX. Olympische Winterspiele) and commonly known as Innsbruck 1964 (Austro-Bavarian: Innschbruck 1964), were a winter multi-sport event which was celebrated in Innsbruck, Austria, from January 29 to February 9, 1964. ... |

Table 21: **Retrieved Wikipedia passage describing generic information about the Olympics, given the question asking about the 4th Summer Olympics.**

| |
|---|
| **[Question]** Who was the Prime Minister of the host country during the 4th edition of the Summer Olympic Games? |
| **[Retrieved Document]** The modern Olympic Games (Olympics; French: Jeux olympiques) are the world's leading international sporting events. They feature summer and winter sports competitions in which thousands of athletes from around the world participate in a variety of competitions. The Olympic Games are considered the world's foremost sports competition, with more than 200 teams, representing sovereign states and territories, participating. ... |

**Robustness analysis of RAG across Temporal Constraint Type** Motivated by the underperformance of RAG observed in Table 7, we conduct a more fine-grained performance analysis, evaluating the robustness of RAG across different temporal constraint types. We measure RAG's success rate in retrieving gold documents (those containing the answer).

Table 22: **RAG's success rate across 13 temporal relations, as defined in Table 10.**

| Performance Level | Temporal Relation (Success rate) |
|---|---|
| High ($> 0.45$) | after (0.67), equal (0.50) |
| Medium (0.45-0.25) | contain (0.33), started-by (0.33), overlapped-by (0.31), start (0.31), finished-by (0.27) |
| Low ($< 0.25$) | before (0.20), met-by (0.17), meet (0.17), finish (0.15), during (0.14), overlap (0.14) |

Table 22 shows two key insights: (1) *Robustness gap across temporal expressions*, where RAG shows 2–4x lower success rates on nuanced operations (e.g., *'overlap'*: 0.14, *'meet'*: 0.17) compared to common operations (e.g., *'equal'*: 0.50, *'after'*: 0.67); (2) *Performance asymmetry between inverse operations*, where performance vary dramatically for semantically opposite pairs – *'before'* (0.20) underperforms *'after'* (0.67) by 3.3x, while some other pairs like *'start'* (0.31) and *'started-by'* (0.33) remain comparable. We believe this kind of analysis can be further applied to diagnose recently developed Temporal RAG strategies (Piryani et al., 2025), such as temporal re-ranking, time-filtered retrieval, or query rewriting.

**Impact of Performance Improvement Techniques on the Single-hop Setting** In addition to the multi-hop setting presented in the main text (Table 7), we provide more results on single-hop questions. Specifically, we apply the same performance improvement techniques to the questions used in the temporal reasoning task (Sec. 4.2) under the closed-book setting to examine the effectiveness of each technique – RAG for providing up-to-date knowledge, and CoT and Time-CoT for improving temporal reasoning – and report the results in Table 23. Interestingly, all techniques greatly improve performances on None-type questions, where we observe that models tend to respond with "no answer" more frequently, possibly rejecting to respond if they are uncertain for their reasoning. In addition, when the additional contexts retrieved via RAG do not contain the gold answer – due to the temporal misalignment issue discussed above – we also observe that models tend to respond with "no answer" rather than relying on their own knowledge, which degrades performances on Multiple- and Unique-type questions while improving performance on None-type questions. As with the multi-hop results, we observe similar performance between CoT and Time-CoT in the single-hop setting.

Table 23: **LLM performances with different performance improvement techniques (RAG, CoT, and Time-CoT) on single-hop questions.** We report performances across question types categorized by answer cardinality: Multiple, Unique, and None, evaluated on the Wikipedia dataset under the closed-book setting. The results are compared with the original setting (Orig.). We report **AT** for Multiple and Unique type questions, and **A** for the None-type questions, as no time reference is expected in no-answer cases.

| Model | Multiple | | | | Unique | | | | None | | | |
|---|---|---|---|---|---|---|---|---|---|---|---|---|
| | Orig. | RAG | CoT | Time-CoT | Orig. | RAG | CoT | Time-CoT | Orig. | RAG | CoT | Time-CoT |
| GPT-3.5 | 12.4 | 14.2 | 9.0 | 0.4 | 38.7 | 26.6 | 26.6 | 12.7 | 56.8 | 54.2 | 83.9 | 97.1 |
| GPT-4 | 20.2 | 25.1 | 21.0 | 30.0 | 61.3 | 54.3 | 65.9 | 57.3 | 19.0 | 39.6 | 51.6 | 74.7 |
| GPT-4o | 34.5 | 30.7 | 24.3 | 32.6 | 65.1 | 59.2 | 62.9 | 62.9 | 37.4 | 43.2 | 60.4 | 50.2 |
| Llama3.1-70B | 20.6 | 8.6 | 16.5 | 19.9 | 59.4 | 34.5 | 68.9 | 48.7 | 22.0 | 54.9 | 25.3 | 33.3 |
| Mixtral-8x7B | 10.9 | 9.7 | 8.6 | 15.4 | 40.6 | 30.3 | 50.9 | 48.7 | 28.9 | 16.5 | 30.8 | 26.7 |
| Gemma2-27B | 24.3 | 13.9 | 19.9 | 21.3 | 59.4 | 45.7 | 62.2 | 62.5 | 6.2 | 16.8 | 10.6 | 12.1 |
| Qwen2-72B | 19.1 | 17.2 | 13.5 | 12.7 | 57.5 | 51.7 | 59.2 | 59.9 | 20.1 | 32.6 | 38.8 | 39.6 |
| Granite3.1-8B | 6.7 | 4.9 | 4.9 | 7.1 | 34.1 | 15.7 | 22.8 | 20.6 | 20.9 | 43.6 | 76.9 | 60.1 |

## D.5 MORE DETAILS ON CORRECTNESS ANALYSIS

Continuing from Sec. 4.4, we provide a more fine-grained correctness analysis result. Similarly to the manual verification setup in Sec. D.2, we manually verify 125 randomly sampled responses per task and measure the correctness using precision, recall, and F1 score. As shown in Table 24, TDBench exhibits high agreement with manual verification, achieving higher precision than recall.

This result indicates that TDBench can effectively capture responses with both correct answers and valid temporal reasoning, while some incorrect explanations – particularly those unrelated to temporal references – remain challenging to detect.

Table 24: **Correctness of TDBench against manual verification across question subcategories**. We use Precision, Recall, and F1 score – see metric definitions in Sec. D.2.

| Task | P | R | F1 |
|---|---|---|---|
| Temporal alignment (Average) | 0.98 | 0.96 | 0.97 |
| Temporal reasoning (Multiple) | 0.85 | 0.88 | 0.86 |
| Temporal reasoning (Unique) | 0.81 | 0.87 | 0.83 |
| Temporal reasoning (None) | 0.89 | 0.90 | 0.90 |
| Temporal reasoning (Average) | 0.87 | 0.91 | 0.88 |
| Multi-hop (Average) | 0.95 | 0.87 | 0.91 |

### D.6 Evaluation on Synthetic Medical Data

Continuing from Sec. 4.5, we demonstrate how TDBench can be applied to synthetic datasets, as long as the underlying tables used in TDBench have TFDs satisfied. We use a synthetic medical dataset consisting of synthetic records of patients at a hospital, and ask questions about doctors and medication by using TFDs *name, gender, blood_type* $\xrightarrow{T}$ *doctor* and *name, gender, blood_type* $\xrightarrow{T}$ *medication* – see example questions in Sec. C.2.

Table 25 shows improved LLMs performances compared to the performances on real-world factual questions, particularly for Multiple- and None-type questions. We hypothesize that this improvement may stem from reduced interference by the model's internal knowledge. As the data is synthetic and unfamiliar, models are more likely to focus on the explicit temporal context provided in the prompt, potentially leading to more accurate temporal reasoning.

Table 25: **LLM performances by answer cardinality evaluated on the synthetic Medical dataset.** Performances are reported across multiple-, unique-, and no-answer ("None") question types under the open-book setting. Only **A** is reported for the None type questions, as no time reference is expected in no-answer cases.

| | Multiple | | | Unique | | | None |
|---|---|---|---|---|---|---|---|
| **Model** | **A** | **T** | **AT** | **A** | **T** | **AT** | **A** |
| GPT-3.5 | 97.5 | 50.4 | 61.9 | 35.4 | 93.6 | 33.0 | **100.0** |
| GPT-4 | **98.8** | 77.9 | 77.7 | 44.4 | **98.6** | 43.9 | 91.0 |
| GPT-4o | 97.5 | **78.6** | 78.5 | **59.8** | 97.2 | **58.4** | **100.0** |
| Llama3.1-70B | 95.8 | 68.6 | **85.8** | 57.9 | 76.6 | 56.4 | 98.8 |
| Mixtral-8x7B | 97.1 | 55.7 | 60.8 | 24.9 | 90.6 | 22.9 | 27.1 |
| Gemma2-27B | 94.8 | 52.1 | 78.3 | 48.3 | 89.8 | 47.0 | **100.0** |
| Qwen2-72B | 93.5 | 68.9 | 66.2 | 53.5 | 95.1 | 52.2 | **100.0** |
| Granite3.1-8B | 88.3 | 44.8 | 41.5 | 17.6 | 93.5 | 17.1 | **100.0** |

### D.7 Evaluation of Task-specific TSQA Methods

Continuing from Sec. 4.5, we provide an additional analysis with task-specific TSQA methods. While the primary goal of our study is to assess the accuracy and reliability of general-purpose models – those expected to possess "accurate knowledge" and widely adopted in many tasks as GPT-Judges for handling factual knowledge (Lin et al., 2021; Sun et al., 2023; Wang et al., 2023) – performance analysis on task-specific models such as TG-LLM (Xiong et al., 2024) and TempT5 (Tan et al., 2023) can offer insights in different perspectives. Specifically, we aim to explore (1) performance differences compared to general-purpose models and (2) the effectiveness of their training methods when measured with other TSQA pairs beyond the original benchmarks used for their training (TGQA (Xiong et al., 2024) and TempREASON (Tan et al., 2023) for TG-LLM and TempT5, respectively), which follow different QA styles.

Table 26: **Performance comparison of task-specific TSQA methods.** Only **A** is reported for TempT5 and T5 due to their answer-only format.

| Model | A | AT |
|---|---|---|
| Llama3.1-70B | 58.4 | 54.2 |
| Mixtral-8x7B | 42.9 | 30.9 |
| Gemma2-27B | 69.3 | 32.8 |
| TG-LLM | 41.0 | 34.0 |
| Llama2-13B (base model of TG-LLM) | 37.0 | 27.0 |
| TempT5 | 4.2 | - |
| T5-base (base model of TempT5) | 0.0 | - |

As shown in Table 26, TG-LLM demonstrates a notable performance improvement over its base model (i.e., Llama2-13B) and performs comparably to the general purpose model Gemma2-27B; however, the performance still lags behind the larger Llama family model used in our experiments (i.e., Llama3.1-70B). In contrast to the good generalization ability shown in TG-LLM, TempT5 greatly struggles with TDBench's QA pairs, resulting in low performance, likely due to the limited capacity of its underlying architecture (i.e., T5-base).

## D.8 COMPARISON WITH LLM-ONLY TSQA METHODS

Continuing from Sec. 3.1 and Sec. 4.5, we provide a comparative study between TDBench and LLM-only TSQA methods. While TDBench leverages both database techniques and an LLM for TSQA construction, one possible approach is to solely use LLMs to directly generate QA pairs from an input data source (Kim et al., 2024; Zhao et al., 2024). We demonstrate how this LLM-only approach can (1) be more prone to hallucinations and (2) incur higher inference costs when processing the raw data.

**Anlaysis on Hallucination**   We demonstrate with Zhao et al. (2024), using their system prompt, shown below. To better reflect realistic multi-attribute reasoning scenarios, we relax one constraint in the original prompt: each question should contain only one column's information.

```
Below is a table in CSV format separated with commas ','. Based on this
table's information along with its description and abstracts,
please raise up to ten questions that the answers are different
in 2010, 2019, and 2023. After raising the question, please get answers
for those time points from the table. Please make sure the answers
should be totally different without overlapping. Please only raise
questions regarding the latest status of those time points instead of
the history, while also do not explicitly mention the time information
in the question to make them more natural. Do not ask questions
cannot be answered based on the information provided in the table. Make
sure the full and explicit names of related entities are used in
the question based on the description in the prompt and do not use
pronouns.
```

We use the same LLM (GPT-4) as in their study and provide 50 rows from the *Country* dataset. The generated QA pairs are as follows:

- $Q_1$: What name was associated with the role of the Prime Minister of Australia?
  $A_1$: Anthony Albanese
- $Q_2$: Who was the Prime Minister of Denmark in 2010?
  $A_2$: Lars Løkke Rasmussen
- $Q_3$: Who was the Prime Minister of Denmark in 2019?
  $A_3$: Lars Løkke Rasmussen
- $Q_4$: Who was serving as the President of Nigeria in the year when Goodluck Jonathan started his term?
  $A_4$: Goodluck Jonathan

- Q$_5$: Who was serving as the Prime Minister of Denmark when Mette Frederiksen began her term?
  A$_5$: Mette Frederiksen

While the model correctly identifies the time-sensitive column based on the table schema (e.g., the name of the role-holder), the generated QA pairs do not fully adhere to the instructions. For instance, (Q$_2$, A$_2$) and (Q$_3$, A$_3$) explicitly mention temporal information, violating the guideline to avoid doing so; (Q$_2$, A$_2$) and (Q$_3$, A$_3$) makes it easy to infer answers from the questions themselves, compromising the question quality.

This case study indicates that LLM-based QA generation can result in less controlled QA generation, being vulnerable to hallucinations even when the provided data source is relatively in small-scale (i.e., 50 rows). In contrast, SQL-based data processing employed in TDBench can ensure more controlled QA generation, even at scale, thereby reducing the manual effort often required in LLM-based methods to identify and eliminate hallucinated responses (Kim et al., 2024; Wu et al., 2024).

**Analysis on Inference Cost**  More importantly, SQL-based processing can be more cost-effective. TDBench only uses the necessary attributes for generating questions via TFDs and discards the rest, whereas an LLM-only approach typically ingests an entire row. We compare TDBench with an LLM-only baseline on five datasets used in our experiments (Sec. 4). As a result, Table xx shows that TDBench uses significantly fewer input tokens while achieving comparable inference time as that of the LLM-only baseline. We also observe greater cost benefits in larger, real-world datasets like the Netflix dataset compared to smaller, well-processed datasets like the Leaders dataset, demonstrating the scalability of TDBench.

Table 27: **Comparison of inference time (Time (s)) and input size (Tokens) across datasets.** #col denotes the column count of each dataset. Bold indicates the lower (better) value.

| Model | Leaders (#col: 5) | | Olympic (#col: 7) | | Environ (#col: 10) | | Medical (#col: 15) | | Netflix (#col: 18) | |
|---|---|---|---|---|---|---|---|---|---|---|
| | *Time (s)* | *Tokens* | *Time (s)* | *Tokens* | *Time (s)* | *Tokens* | *Time (s)* | *Tokens* | *Time (s)* | *Tokens* |
| LLM-only | 1.79 | 206 | **1.89** | 316 | **1.47** | 376 | **1.81** | 542 | 1.91 | 1,288 |
| TDBench | **1.62** | **161** | 2.57 | **177** | 1.93 | **169** | 3.32 | **299** | **1.73** | **138** |

D.9  CROSS-TRANSLATOR SIMILARITY ANALYSIS

Continuing from Sec. 4.5, we provide a cross-translator similarity analysis to assess the impact of the translator model used in the QA construction process. When using an LLM as a SQL-to-text translator, the generated questions may implicitly reflect biases such as question-style familiarity, which can affect evaluation when the same model is also an evaluation target—for example, we use GPT-4o as the SQL-to-text translator, and GPT-4o is also one of the models being evaluated. However, we emphasize that, rather than relying on free-form prompting, all questions are generated from the same underlying SQL queries, which constrains the generation across translator models and limits the potential bias introduced by the translator.

To validate that different translators produce similar questions, we additionally conducted a cross-generator similarity analysis. We compared questions generated from different translator models with GPT-4o-generated questions by inputting 100 randomly sampled SQL queries used in TDBench. We measurie similarities in two complementary dimensions via widely-adopted metrics: (1) *semantic similarity* using sentence embeddings (Reimers & Gurevych, 2019) (all-mpnet-base-v2), which captures whether questions convey equivalent meaning regardless of exact wording; and (2) *lexical similarity* using ROUGE-1 F1 score (Lin, 2004), which measures surface-level unigram overlap.

The results shown in Table 28 demonstrate very high similarity across translators (semantic similarity > 0.85, ROUGE-1 F1 > 0.65), indicating that SQL constraints dominate generation and suggest minimal impact from the translator model dual role. However, users should note that these metrics may not capture all subtle biases in question formulation and can freely choose the translator model based on their own constraints and evaluation targets.

Table 28: **Cross-generator similarity analysis across different SQL-to-text translator models.**

| Metric | Llama3.1-70B | Mixtral-8x7B | Qwen2-72B | Gemma2-27B |
|---|---|---|---|---|
| Semantic Similarity | $0.942 \pm 0.042$ | $0.921 \pm 0.052$ | $\mathbf{0.945 \pm 0.035}$ | $0.930 \pm 0.051$ |
| ROUGE-1 F1 | $0.781 \pm 0.147$ | $0.673 \pm 0.080$ | $\mathbf{0.810 \pm 0.094}$ | $0.717 \pm 0.113$ |

### D.10 FULL EXPERIMENTAL RESULTS

Continuing from Sec. 4.3, we provide full results on the multi-hop setting. Table 29 shows that Llama3.1-70B outperforms GPT-4o in the 2-hop setting, whereas GPT-4o outperforms Llama3.1-70B in the 3-hop setting. Interestingly, GPT-4o shows an extremely low hallucination rate (i.e., $H_1$) in the 2-hop setting – which requires identifying the role-holder of the host country of a given Olympic games – indicating that once the model get the correct host country and time, the model is mostly able to retrieve the correct role-holder for that time.

Table 29: **Full results for performance comparison on implicit, multi-hop questions.** $H_i$ denotes the hallucination rate at the $(i + 1)$-st hop given that the $i$-th hop is correct.

| Model | 2-hop | | | 3-hop | | | |
|---|---|---|---|---|---|---|---|
| | **A** | **AT** | $\mathbf{H_1}$ | **A** | **AT** | $\mathbf{H_1}$ | $\mathbf{H_2}$ |
| GPT-3.5 | 66.7 | 63.1 | 9.1 | 33.8 | 33.8 | **8.5** | 7.6 |
| GPT-4 | 88.4 | 67.2 | 4.5 | 57.1 | 52.5 | 10.8 | 0.5 |
| GPT-4o | 86.9 | 79.3 | **0.0** | **90.4** | **87.9** | 26.3 | **0.0** |
| Llama3.1-70B | **91.9** | **83.8** | 68.8 | 82.8 | 80.8 | 38.2 | 4.5 |
| Mixtral-8x7B | 80.8 | 72.7 | 45.9 | 60.1 | 49.0 | 19.5 | 10.6 |
| Gemma2-27B | 79.8 | 77.8 | 35.1 | 48.0 | 43.9 | 26.5 | 24.7 |
| Qwen2-72B | 78.3 | 65.2 | 50.0 | 68.2 | 59.1 | 35.1 | 5.6 |
| Granite3.1-8B | 73.7 | 50.5 | 80.4 | 11.1 | 11.1 | 10.3 | 77.8 |

