# OpenReview forum: "Harnessing Temporal Databases for Systematic Evaluation of Factual Time-Sensitive Question-Answering in LLMs"
_ICLR.cc/2026/Conference — ICLR 2026 Poster_

### Official Review · Reviewer_wK6u · 2025-10-30

**Soundness:** 3
**Presentation:** 3
**Contribution:** 3
**Rating:** 6
**Confidence:** 3

**Summary:**

This paper tackles a key problem: creating benchmarks for time-sensitive QA is slow and manual. The proposedTDBench uses temporal databases and SQL to automatically generate diverse and reliable QA pairs, bypassing the usual human effort. A key contribution is the *Time Accuracy* metric, which goes beyond checking the final answer to also verify the dates in the model's explanation. This provides a much deeper look into model factuality and hallucination.

**Strengths:**

- The core idea of using temporal database theory to systematically generate a benchmark is highly innovative and effective. It is a bridge between two fields.
- TDBench automates a tedious, expensive process. This makes it highly practical for both academic research and industry applications where up-to-date, domain-specific evaluation is needed.
- Time accuracy reveals a critical layer of hallucination—correct answers with faulty temporal reasoning—that is otherwise missed by standard benchmarks.
- The framework's use of 13 temporal relations ensures a much broader and more rigorous test of LLM capabilities than existing template-based methods.

**Weaknesses:**

- The framework's success hinges on having a well-structured temporal database. This shifts the manual effort from question creation to data modeling and curation, which can be a significant bottleneck in itself.
- The paper notes a ~9% error rate in the SQL-to-text step. This introduces a small but non-negligible risk that models are being evaluated on slightly flawed or ambiguous questions.
- The need to explicitly prompt models for date information to measure *Time Accuracy*, while necessary, creates a test condition that may not fully reflect natural user interaction.

**Questions:**

The framework elegantly shifts the manual effort from question design to database design. How do you see TDBench being applied in settings where the source data is semi-structured or messy (e.g., tables scraped from the web)? Does the automation promise hold if a significant data curation phase is required first?

---

> ### Author Response · Authors · 2025-11-21
>
> To Reviewer wK6u (Response 1/2),
>
> We appreciate your constructive feedback. We respond to each of your points below.
>
>
> > **[W1]** *The framework's success hinges on having a well-structured temporal database. This shifts the manual effort from question creation to data modeling and curation, which can be a significant bottleneck in itself*.
>
> We value your perspective, but would like to clarify two aspects below.
>
> **(1) Temporal databases are widespread, not rare**
>
> Contrary to your concern, **structured temporal data is prevalent across many critical domains**:
> - **Healthcare**: Electronic Health Records (EHRs), patient timelines, treatment records [1, 2]
> - **Finance**: Trading records, transaction histories, regulatory data [3, 4]
> - **Enterprise**: Supply chain, Customer Relationship Management (CRM), Human Resources (HR) databases  [5]
>
> These databases often serve operational needs, making TDBench directly applicable without additional data creation effort. Public platforms like Kaggle [6] and government data portals [7] also provide readily accessible datasets across diverse domains – Legal, Environmental, Cultural, and Social – as demonstrated in our experiments.
>
>
> **(2) Wikidata/Wikipedia do not represent all data**
>
> While LLMs are increasingly applied to specialized domains – such as healthcare [8] or enterprise analytics [9] – **these domains often fall beyond Wikipedia/Wikidata due to privacy and proprietary restrictions** [10]. TDBench enables the use of existing temporal databases in these domains for domain-specific benchmarking, offering a practical and complementary tool to current Wikidata/Wikipedia-dominant TSQA benchmarking [11, 12, 13, 14, 15].
>
> Therefore, while you correctly identify that **TDBench follows data-centric TSQA benchmarking [11, 13, 14, 16, 17],  which depends on data availability, TDBench substantially broadens the scope of “usable datasets”** by incorporating structured tabular data. To reflect your valuable feedback, we **have added the above discussions on this data-centric aspect in our revision (Sec. 7 and Sec. A.1, highlighted in blue)**.
>
>
>
> &nbsp;
>
> > **[W2]** *The paper notes a ~9% error rate in the SQL-to-text step. This introduces a small but non-negligible risk that models are being evaluated on slightly flawed or ambiguous questions*.
>
> We appreciate your constructive feedback. As you said, our SQL-to-text step is not entirely free of hallucinations – as it still involves an LLM – however, **integration with database techniques substantially reduces hallucination and ambiguity risks compared to LLM-only QA generation**:
>
> **(1) More controlled data processing via SQL execution**
>
> TDBench’s SQL-based data processing provides a more stable approach than relying solely on free-form LLM prompting. As we demonstrate in Sec. D.8, directly feeding text to an LLM often yields undesired and inconsistent outputs – out of 10 pairs, 2 pairs not conveying the instruction for time-related QA and 1 pair having a hallucinated answer – which occurred even on very small databases (<100 rows). This inconsistency would only worsen at larger real-world datasets, making TDBench’s SQL-based approach increasingly beneficial.
>
>
> **(2) Guaranteed generation of time-related questions via TFDs**
>
> Even when an LLM generates a valid QA pair, the question may not involve temporal reasoning, making it unsuitable for TSQA – an issue commonly reported in LLM-only TSQA benchmarks [16, 18]. Our use of TFDs explicitly restricts generation to time-relevant attributes, effectively reducing such ambiguity and the need for manual filtering [16, 18].
>
> While combining database techniques with LLMs in TDBench provides a meaningful advantage in reducing flawed or ambiguous questions, we agree that further reducing errors is always desirable. We respectfully note that a detailed categorization of SQL-to-Text error types is provided in Sec. B.3 for transparency.
>
>
>
>
>
> &nbsp;
>
> > **[W3]** *The need to explicitly prompt models for date information to measure Time Accuracy, while necessary, creates a test condition that may not fully reflect natural user interaction*.
>
> **As a benchmarking paper, we prioritized ensuring fair and consistent comparison across models.** Without explicit prompts, models may only provide the final answer and bypass the intended temporal reasoning evaluation. Thus, while not fully reflecting natural user interaction as you said, we intentionally used explicit prompting to prevent such cases and to evaluate models under the same conditions.

---

> ### Author Response · Authors · 2025-11-21
>
> To Reviewer wK6u (Response 2/2),
>
> > **[Q1]** *The framework elegantly shifts the manual effort from question design to database design. How do you see TDBench being applied in settings where the source data is semi-structured or messy (e.g., tables scraped from the web)? Does the automation promise hold if a significant data curation phase is required first?*
>
> We appreciate your great questions and respond to each point below.
>
> **(1) Extension to semi-structured data**
>
> Although not our primary focus, TDBench can be **readily applied to some semi-structured data by transforming this data into structured formats**. For example, techniques like schema inference and normalization [19, 20] can be utilized during the data preprocessing stage.
>
> **(2) Handling noisy data**
>
> Data quality issues – such as factual inconsistencies and domain-specific anomalies [21]  – can affect TDBench and **lead to noisy QA pairs, which is a challenge common to all data-centric benchmarking approaches** [11, 13, 14, 16, 17]. While there is extensive work on enhancing data quality [22, 23, 24], our focus is orthogonal, where we enhance the downstream pipeline to best leverage the available data. Thus, while TDBench provides clear benefits in this context – providing a generalizable and effective tool for converting one’s data into a benchmark – integration with existing data cleaning techniques [9, 10] would further strengthen our contribution.
>
> **(3) Scope of automation**
>
> Continuing from (2), **we thus do not consider data curation as our automation scope**, but rather assume the available data. We note that temporal databases are widespread in many critical domains, as we discussed in W1.
>
> **To reflect all your questions, we have added discussions on (1) extension to semi-structured data (Sec. B.2), (2) data-centric aspect and potential risks of noisy data (Sec. 7), and (3) availability of temporal databases across diverse domains (Sec. A.1).** All modifications are highlighted in blue.
>
> &nbsp;
>
> We hope our response addresses your questions. We also do appreciate your acknowledgement of our automatic framework design with database techniques and the necessity of time accuracy metric. Please let us know if you have any remaining questions.
>
> &nbsp;
>
> > **References**
>
> [1] Johnson et al., MIMIC-III, a freely accessible critical care database, Nature Scientific Data 2016.
> [2] Pollard et al., The eICU Collaborative Research Database, a freely available multi-center database for critical care research, Nature Scientific Data 2018.
> [3] Zheng et al., Global table extractor (gte): A framework for joint table identification and cell structure recognition using visual context, WACV 2021.
> [4] Zhu et al., TAT-QA: A question answering benchmark on a hybrid of tabular and textual content in finance, ACL 2021.
> [5] Jensen et al., Temporal data management, IEEE TKDE 1999.
> [6] Prasad Patil, Healthcare dataset, Kaggle 2025, https://www.kaggle.com/datasets/prasad22/healthcare-dataset.
> [7] World Bank Group, State and Trends of Carbon Pricing Dashboard 2025, https://carbonpricingdashboard.worldbank.org/compliance/instrument-detail.
> [8] He et al., Survey of large language models for healthcare: from data, technology, and applications to accountability and ethics, Information Fusion 2024.
> [9] Weng et al., DataLab: A unified platform for LLM-powered business intelligence, ACM SIGKDD 2024.
> [10] Regulation (EU) 2016/679 of the European Parliament and of the Council, 2016.
> [11] Jia et al., Tempquestions: A benchmark for temporal question answering, WWW 2018.
> [12] Chen et al., A dataset for answering time-sensitive questions, NeurIPS 2021.
> [13] Jang et al., Temporalwiki: A lifelong benchmark for training and evaluating ever-evolving language models, EMNLP 2022.
> [14] Gupta et al., Temptabqa: Temporal question answering for semi-structured tables, EMNLP 2023.
> [15] Tan et al., Towards benchmarking and improving the temporal reasoning capability of large language models, ACL 2023.
> [16] Kim et al., Carpe diem: On the evaluation of world knowledge in lifelong language models, NAACL 2024.
> [17] Wu et al., AntiLeakBench: Preventing data contamination by automatically constructing benchmarks with updated real-world knowledge, ACL 2025.
> [18] Zhao et al., Set the clock: Temporal alignment of pretrained language models, ACL Findings 2024.
> [19] DiScala et al., Automatic generation of normalized relational schemas from nested key-value data, SIGMOD 2016.
> [20] Baazizi et al., Parametric schema inference for massive JSON datasets, VLDB 2019.
> [21] Wang et al., Beyond accuracy: What data quality means to data consumers, JISTOR 1996.
> [22] Batini et al., Methodologies for data quality assessment and improvement, ACL CSUR 2009.
> [23] Chu et al., Data cleaning: Overview and emerging challenges, ACM Books 2016.
> [24] Rekatsinas et al., Holoclean: Holistic data repairs with probabilistic inference, VLDB 2017.

---

### Official Review · Reviewer_YGnM · 2025-11-01

**Soundness:** 3
**Presentation:** 3
**Contribution:** 2
**Rating:** 4
**Confidence:** 4

**Summary:**

This paper presents the TDBench framework to evaluate LLMs on time-sensitive QA tasks. The framework addresses manual bottlenecks in existing TSQA benchmarks by automating both the construction and evaluation processes. TDBench uses Temporal Functional Dependencies (TFDs) to automatically identify valid QA attribute pairs. Temporal SQLs are generated attributes identified and 13 temporal relations. These SQLs are now translated into Natural Language Questions using GPT-4o, and the ground-truth answer is obtained by executing the query against the database. A new metric, "Time Accuracy", is also introduced, which evaluates the correctness of the temporal references in a generated answer.

**Strengths:**

1. Automated and Scalable QA construction. Unlike prior works, which use manual curation or templates, TDBench automates this process with TFDs.
2. TDBench can be extended to any database with temporal data.

**Weaknesses:**

1. The paper argues that it eliminates the manual bottleneck of benchmark creation. Well-structured temporal databases with defined TFDs for many domains may not exist, and it's not an easy task to automate their creation.
2. In many real-world databases, schemas are updated with time as well. Even in Wikipedia, many infobox keys & values are constantly being updated. This is a major limitation for TDBench like tasks.
3. While the time accuracy metric is good, it's not anything new.

**Questions:**

1. Can you provide guidance or tools for automatically discovering TFDs from existing databases?

---

> ### Author Response · Authors · 2025-11-21
>
> To Reviewer YGnM (Response 1/2),
>
> We appreciate your constructive feedback. We respond to each of your points below.
>
> > **[W1]** *The paper argues that it eliminates the manual bottleneck of benchmark creation. Well-structured temporal databases with defined TFDs for many domains may not exist, and it's not an easy task to automate their creation*.
>
> We value your perspective, but would like to clarify two aspects below.
>
> **(1) Temporal databases are widespread, not rare**
>
> Contrary to your concern, **structured temporal data is prevalent across many critical domains**:
> - **Healthcare**: Electronic Health Records (EHRs), patient timelines, treatment records [1, 2]
> - **Finance**: Trading records, transaction histories, regulatory data [3, 4]
> - **Enterprise**: Supply chain, Customer Relationship Management (CRM), Human Resources (HR) databases  [5]
>
> These databases often serve operational needs, making TDBench directly applicable without additional data creation effort. Public platforms like Kaggle [6] and government data portals [7] also provide readily accessible datasets across diverse domains – Legal, Environmental, Cultural, and Social – as demonstrated in our experiments.
>
> **(2) Wikidata/Wikipedia do not represent all data**
>
> While LLMs are increasingly applied to specialized domains – such as healthcare [8] or enterprise analytics [9] – **these domains often fall beyond Wikipedia/Wikidata due to privacy and proprietary restrictions** [10]. TDBench enables the use of existing temporal databases in these domains for domain-specific benchmarking, offering a practical and complementary tool to current Wikidata/Wikipedia-dominant TSQA benchmarking [11, 12, 13, 14, 15].
>
> Therefore, while you correctly identify that **TDBench follows data-centric TSQA benchmarking [11, 13, 14, 16, 17],  which depends on data availability, TDBench substantially broadens the scope of “usable datasets”** by incorporating structured tabular data. To reflect your valuable feedback, **we have added the above discussions on this data-centric aspect in our revision (Sec. 7 and Sec.A.1, highlighted in blue)**.
>
> &nbsp;
>
> > **[W2]** *In many real-world databases, schemas are updated with time as well. Even in Wikipedia, many infobox keys & values are constantly being updated. This is a major limitation for TDBench like tasks*.
>
> Thank you for the great comment. Including schema evolution, real-world data evolves over time – and **this is precisely why we design TDBench as a dynamic benchmarking framework**.
>
> When the underlying database schema changes, **users simply need to update the corresponding TFD**, and TDBench will automatically regenerate the benchmark to align with the new schema. Thus, real-world dynamics such as schema evolution indicate not a limitation, but rather **a significant advantage of TDBench compared to static benchmarks  [12, 14, 18, 19, 20, 21]**, which require manual re-annotation or template redesign whenever the schema changes.
>
> &nbsp;
>
> > **[W3]** *While the time accuracy metric is good, it's not anything new*.
>
> We respectfully emphasize that **our contribution lies not in the metric itself, but in the valuable insights gained from the time accuracy metric**. Using this metric, we conduct the first systematic diagnosis of temporal reasoning quality in TSQA, which has been largely neglected by answer-only evaluation in current TSQA research.
>
> More specifically, we newly find:
> - **Substantial gaps between answer accuracy and time accuracy** (on average 21% in our experiments), indicating risk of overestimating TSQA performance.
> - **Ranking reversals among LLMs that look similar under answer accuracy**, for example, in Table 5 LLaMA3.1-70B and Qwen2-72B have comparable answer accuracy, yet Qwen2-72B significantly outperforms LLaMA3.1-70B on time accuracy.
>
>
> **These findings demonstrate how time accuracy can unlock many new insights, which may also benefit future TSQA research**. We also hope our motivation for time accuracy – a model should not only provide correct answers, but also correct temporal reasoning – can further facilitate the development of trustworthy LLMs via a more comprehensive TSQA evaluation.
>
> &nbsp;

---

> ### Author Response · Authors · 2025-11-21
>
> To Reviewer YGnM (Response 2/2),
>
> > **[Q1]** *Can you provide guidance or tools for automatically discovering TFDs from existing databases*?
>
> We appreciate your insightful question. **There are several existing algorithms and tools for automatically discovering FD/TFDs**, including algorithms such as Dep-Miner [22], FastFDs [23], and DFD [24], and a Python library called FDTool [24].
>
> Based on your great question, **we have included a discussion on these readily available tools in our revision (Sec. B.2, highlighted in blue)** to assist users in handling unfamiliar databases. **We will also provide guidance on the FDTool Python library** in our open-source GitHub repository.
>
> &nbsp;
>
> We hope our response addresses your questions. We also do appreciate your acknowledgement of our framework design with database techniques and the value of generalization to arbitrary domains. Please let us know if you have any remaining questions.
>
> &nbsp;
>
> > **References**
>
> [1] Johnson et al., MIMIC-III, a freely accessible critical care database, Nature Scientific Data 2016.
> [2] Pollard et al., The eICU Collaborative Research Database, a freely available multi-center database for critical care research, Nature Scientific Data 2018.
> [3] Zheng et al., Global table extractor (gte): A framework for joint table identification and cell structure recognition using visual context, WACV 2021.
> [4] Zhu et al., TAT-QA: A question answering benchmark on a hybrid of tabular and textual content in finance, ACL 2021.
> [5] Jensen et al., Temporal data management, IEEE TKDE 1999.
> [6] Prasad Patil, Healthcare dataset, Kaggle 2025, https://www.kaggle.com/datasets/prasad22/healthcare-dataset.
> [7] World Bank Group, State and Trends of Carbon Pricing Dashboard 2025, https://carbonpricingdashboard.worldbank.org/compliance/instrument-detail.
> [8] He et al., Survey of large language models for healthcare: from data, technology, and applications to accountability and ethics, Information Fusion 2024.
> [9] Weng et al., DataLab: A unified platform for LLM-powered business intelligence, ACM SIGKDD 2024.
> [10] Regulation (EU) 2016/679 of the European Parliament and of the Council, 2016.
> [11] Jia et al., Tempquestions: A benchmark for temporal question answering, WWW 2018.
> [12] Chen et al., A dataset for answering time-sensitive questions, NeurIPS 2021.
> [13] Jang et al., Temporalwiki: A lifelong benchmark for training and evaluating ever-evolving language models, EMNLP 2022.
> [14] Gupta et al., Temptabqa: Temporal question answering for semi-structured tables, EMNLP 2023.
> [15] Tan et al., Towards benchmarking and improving the temporal reasoning capability of large language models, ACL 2023.
> [16] Kim et al., Carpe diem: On the evaluation of world knowledge in lifelong language models, NAACL 2024.
> [17] Wu et al., AntiLeakBench: Preventing data contamination by automatically constructing benchmarks with updated real-world knowledge, ACL 2025.
> [18] Wei et al., Menatqa: A new dataset for testing the temporal comprehension and reasoning abilities of large language models, EMNLP Findings 2023.
> [19] Kasai et al., Realtime qa: What's the answer right now?, NeurIPS 2023.
> [20] Vu et al. Freshllms: Refreshing large language models with search engine augmentation, ACL Findings 2024.
> [21] Jia et al., Tiq: A benchmark for temporal question answering with implicit time constraints, WWW 2025.
> [22] Lopes et al., Efficient discovery of functional dependencies and Armstrong relations, EDBT 2000.
> [23] Agrawal et al., Mining association rules between sets of items in large databases, ACM SIGMOD 1993.
> [24] Abedjan et al., DFD: Efficient functional dependency discovery, PVLDB 2014.
> [25] Buranosky et al., FDTool: a Python application to mine for functional dependencies and candidate keys in tabular data, PubMed 2019.

---

> ### Comment · Reviewer_YGnM · 2025-11-24
> **Thanks for your response!**
>
> Thanks for your detailed response. The additional insights you provided have successfully addressed my concerns. As a result, I am happy to raise my score for the submission.

---

### Official Review · Reviewer_AktZ · 2025-11-01

**Soundness:** 3
**Presentation:** 3
**Contribution:** 3
**Rating:** 4
**Confidence:** 4

**Summary:**

This paper proposes TDBench, a benchmark for time-sensitive factual question answering. The benchmark leverages three database techniques to construct questions directly from temporal databases: using Temporal Functional Dependencies to select answerable facts, employing temporal SQL to encode diverse temporal contexts, and utilizing temporal joins to create multi-hop event-event reasoning items. Additionally, the paper introduces a new evaluation metric, "time accuracy," which verifies whether a model's explanation cites correct temporal evidence in addition to providing the correct answer. Experiments on multiple large language models reveal a significant gap between answer accuracy and answer-plus-time accuracy.

**Strengths:**

1. The use of Temporal Functional Dependencies  for knowledge selection and temporal SQL based on all 13 Allen relations to generate questions is rigorous, systematic, and scalable. This enables the creation of more diverse and complex temporal reasoning questions than those commonly found in existing benchmarks.

2. The introduction of the "time accuracy" concept correctly emphasizes that in TSQA, the rationale is as important as the answer itself.

3. By building on fundamental database principles, TDBench is not tied to specific knowledge sources. The authors demonstrate this advantage by evaluating LLMs on domain-specific datasets, revealing performance variations overlooked by Wikipedia-centric benchmarks. This makes the framework highly valuable for creating application-specific evaluations.

**Weaknesses:**

1. The accuracy of LLM generation and evaluation is relatively low, with known misjudgment issues. This may introduce label noise and lead to benchmark bias. It is recommended to incorporate more automated verification or manual validation mechanisms.

2. Temporal information in many scenarios includes open/uncertain intervals or requires granularities beyond dates (e.g., partial months, fiscal quarters). The current time accuracy standard, which only validates the start/end fields of each relation, is relatively limited in many contexts.

3. The same provider's model (GPT-4o) is used in the pipeline (SQL-to-text) and is also an evaluation target. Although the authors avoid using GPT-4o as a judge to reduce bias, this dual role still requires further analysis.

4. Many existing studies on TSQA dataset construction are not effectively cited or analyzed. Beyond template-based approaches, other methods such as manual construction and LLM-automated construction [1][2][3] should be included and discussed in the related work section.

[1] A Question Answering Dataset for Temporal-Sensitive Retrieval-Augmented Generation

[2] SituatedQA: Incorporating Extra-Linguistic Contexts into QA

[3] ComplexTempQA: A 100m Dataset for Complex Temporal Question Answering

**Questions:**

None

---

> ### Author Response · Authors · 2025-11-21
>
> To Reviewer AktZ (Response 1/2),
>
> We appreciate your constructive feedback. We respond to each of your points below.
>
> > **[W1]** *The accuracy of LLM generation and evaluation is relatively low, with known misjudgment issues. This may introduce label noise and lead to benchmark bias. It is recommended to incorporate more automated verification or manual validation mechanisms*.
>
> We do agree that LLM-only benchmarking can be prone to hallucinations, leading to low accuracy – and **this is precisely why we designed TDBench with SQL-based grounding**.
>
> **Reducing label noise via SQL-generated answers:**
>
> LLMs can hallucinate during QA generation, where hallucination on answers creates incorrect labels that corrupt the benchmark. We provide a detailed example in Sec. D.8, where an LLM generates inconsistent answers given the same questions. In contrast, TDBench generates answers through SQL execution, ensuring that the answers are grounded in the database and significantly reducing label noise due to LLM hallucination.
>
> **Mitigating benchmark bias via SQL-oriented temporal expressions:**
>
> LLMs are known to exhibit repetitive patterns and limited diversity in generation [1, 2] that can create benchmark bias. For instance, LLMs might generate only simple temporal expressions (e.g., "in 2020") while neglecting complex temporal relationships (e.g., "meets", "overlaps"). In contrast, TDBench algorithmically generates diverse temporal expressions based on Allen's interval algebra [3], guaranteeing coverage on temporal relation types and sidestepping the potential risk of repetitive patterns.
>
> &nbsp;
>
> > **[W2]** *Temporal information in many scenarios includes open/uncertain intervals or requires granularities beyond dates (e.g., partial months, fiscal quarters). The current time accuracy standard, which only validates the start/end fields of each relation, is relatively limited in many contexts*.
>
> We appreciate your insightful comment. While you correctly identify that TDBench does not consider non-standard temporal units (e.g., fiscal quarters) as first-class representations, **TDBench is highly extensible as follows**:
>
> - **Open intervals** using null formats (e.g., ``[2025-11-14, NULL]``), as utilized in generating temporal alignment questions
> - **Granularities beyond date** using a mapping to a set of time ranges (e.g., FY2020 Q1 => ``[2020-01-01, 2020-03-31]``, FY2020 Q2 => ``[2020-04-01, 2020-06-30]`` for fiscal quarters, and “early June 2020” => ``[2020-06-01, 2020-06-10]`` for partial months)
> - **Finer-grained time representation** using database data types (e.g.,  ``2020-01-01 13:10:01`` via ``TIMESTAMP`` type)
>
> In addition, **updating the benchmark to incorporate such granularities is easy for TDBench**, where only database-level preprocessing is required, and TDBench will automatically regenerate the benchmark. This dynamic nature offers a substantial advantage compared to static benchmarks [4, 5, 6, 7, 8, 9] -- which would need manual re-annotation for each fiscal quarter variant.
>
> We also agree that extending to a more nuanced temporal information such as uncertain intervals is a valuable future direction. **Reflecting your insightful comment, we added this discussion as a potential extension to non-standard temporal units in our revision (Sec. B.5, highlighted in blue)**.
>
> &nbsp;

---

> ### Author Response · Authors · 2025-11-21
>
> To Reviewer AktZ (Response 2/2),
>
> > **[W3]** *The same provider's model (GPT-4o) is used in the pipeline (SQL-to-text) and is also an evaluation target. Although the authors avoid using GPT-4o as a judge to reduce bias, this dual role still requires further analysis*.
>
> Thank you for your valuable observation. While we did avoid using GPT-4o as a judge that can incur significant evaluation bias, using GPT-4o as a SQL translator may still implicitly introduce bias in the generated questions. However, rather than free-form prompting, questions are generated from the same SQL queries – which **constrains generation across translator models and limits potential bias from GPT-4o's involvement**.
>
> **To validate that different translators produce similar questions, we additionally conducted a cross-generator similarity analysis.** We compared questions generated from different translator models by inputting 100 randomly sampled SQL queries used in TDBench, measuring similarities in two complementary dimensions via widely-adopted metrics [10, 11]:
> -  *Semantic similarity* using sentence embeddings [12] (all-mpnet-base-v2), which captures whether questions convey equivalent meaning regardless of exact wording.
> - *Lexical similarity* using ROUGE-1 F1 score [13], which measures surface-level unigram overlap.
>
>
> **The results demonstrate very high similarity across translators** (semantic similarity > 0.85 [12, 14], ROUGE-1 F1 > 0.65 [13, 15]), indicating that SQL constraints dominate generation and suggest minimal impact from GPT-4o's dual role.
>
> | Metric              | LLama-3.1-70B      | Mixtral 8x7B       | Qwen2-72B         | Gemma2-27B        |
> |--|---|----|--|--|
> | Semantic Similarity | 0.942 ± 0.042      | 0.921 ± 0.052       | 0.945 ± 0.035     | 0.930 ± 0.051     |
> | ROUGE-1 F1          | 0.781 ± 0.147      | 0.673 ± 0.080       | 0.810 ± 0.094     | 0.717 ± 0.113     |
>
> Nevertheless, we respect your perspective where subtle biases beyond what these metrics capture may still exist. To help users make informed decisions, **we have added guidance on potential translator biases and the above cross-generator analysis in our revision (Sec. B.3, highlighted in blue)**, including the option to use alternative translator models beyond GPT-4o.
>
>
>
> &nbsp;
>
> > **[W4]** *Many existing studies on TSQA dataset construction are not effectively cited or analyzed. Beyond template-based approaches, other methods such as manual construction and LLM-automated construction. [16] A Question Answering Dataset for Temporal-Sensitive Retrieval-Augmented Generation [17] SituatedQA: Incorporating Extra-Linguistic Contexts into QA [18] ComplexTempQA: A 100m Dataset for Complex Temporal Question Answering*
>
>
> We appreciate your suggested papers and **have added the citations in our revision (Sec. 5, highlighted in blue)**; note that [18] was published after our submission date. **We respectfully note that TDBench is already compared against the categories represented by these works** – LLM-automated approaches (including [16]), manually constructed benchmarks (including [17]), as well as template-based approaches (including [18]).
>
> &nbsp;
>
> We hope our response addresses your questions. We also do appreciate your acknowledgement of our framework design with database techniques, the necessity of the time accuracy metric, and the value of generalization to domain-specific datasets. Please let us know if you have any remaining questions.

---

> ### Author Response · Authors · 2025-11-21
>
> > **References**
>
> [1] Holtzman et al., The curious case of neural text degeneration, ICLR 2020.
> [2] Welleck et al., Neural text generation with unlikelihood training, ICLR 2020.
> [3] Allen et al., Maintaining knowledge about temporal intervals, ACM Comm 1983.
> [4] Chen et al., A dataset for answering time-sensitive questions, NeurIPS 2021.
> [5] Wei et al., Menatqa: A new dataset for testing the temporal comprehension and reasoning abilities of large language models, EMNLP Findings 2023.
> [6] Kasai et al., Realtime qa: What's the answer right now?, NeurIPS 2023.
> [7] Gupta et al., Temptabqa: Temporal question answering for semi-structured tables, EMNLP 2023.
> [8] Vu et al. Freshllms: Refreshing large language models with search engine augmentation, ACL Findings 2024.
> [9] Jia et al., Tiq: A benchmark for temporal question answering with implicit time constraints, WWW 2025.
> [10] Celikyilmaz et al., Evaluation of text generation: A survey, arXiv 2020.
> [11] Marie et al., Scientific credibility of machine translation research: A meta-evaluation of 769 papers, ACL 2021.
> [12] Reimers et al., Sentence-BERT: Sentence embeddings using Siamese BERT-networks, EMNLP 2019.
> [13] Lin et al., Rouge: A package for automatic evaluation of summaries, ACL 2004.
> [14] Cer et al., Semeval-2017 task 1: Semantic textual similarity-multilingual and cross-lingual focused evaluation, ACL SemEval 2017.
> [15] Bandel et al., Quality controlled paraphrase generation, ACL 2022.

---

> ### Comment · Reviewer_AktZ · 2025-11-26
> **response**
>
> I thank the authors for their response. They have addressed my concerns, and I have updated my review score accordingly.

---

### Official Review · Reviewer_fr2Z · 2025-11-03

**Soundness:** 3
**Presentation:** 4
**Contribution:** 3
**Rating:** 6
**Confidence:** 4

**Summary:**

The paper introduces TDBench, a framework that auto-constructs time-sensitive QA (TSQA) benchmarks from temporal databases using temporal functional dependencies (TFDs), temporal SQL over the full set of 13 temporal relations, and SQL-to-text generation, then evaluates models on both answer accuracy and a proposed time-accuracy metric that checks the correctness of referenced timestamps in explanations. The pipeline (knowledge selection via TFDs → temporal SQL query generation → SQL-to-text question conversion) yields scalable benchmarks spanning Wikipedia-like and domain-specific datasets (e.g., law, carbon tax, UNESCO, Netflix). Experiments on 8 LLMs show a sizable gap between answer-only and answer-plus-time evaluation, and analyses cover relation types, answer cardinality (multiple/unique/none), temporal spans, and multi-hop (via temporal joins). The authors report high agreement with manual verification and argue TDBench generalizes beyond Wikidata/Wikipedia and reduces manual curation burden.

**Strengths:**

* The paper is well-positioned relative to prior work. It offers clear contrast to template-based and Wikidata-constrained temporal QA benchmarks by covering a wider range of relations and incorporating explanation evaluation, an element often absent from earlier efforts.

* The proposed framework is clear and modular. It employs TFDs for target selection, leverages temporal SQL to ensure comprehensive coverage of 13 Allen relations, and uses SQL-to-text conversion to generate natural questions. This approach is both principled and generalizable across data schemas.

* The evaluation extends meaningfully beyond answer accuracy. The introduction of a time-accuracy metric enables explanation-level verification, revealing cases where models identify correct entities but hallucinate timestamps, which results in a notable 21.7% average performance drop.

* The empirical analysis is comprehensive. The authors examine relation-wise performance (e.g., strong on equal, weaker on contain or overlap), answer cardinality (including “none”), temporal span effects, and multi-hop reasoning with hop-wise hallucination (H1/H2), providing fine-grained diagnostic insights.

**Weaknesses:**

* The paper lacks clarity in how “current” time is grounded. Temporal-alignment questions depend on explicit date references (e.g., “as of 2025”), yet it remains unclear how “now” is defined, stored, and exposed across datasets to ensure reproducibility and longitudinal comparability. Clearer date-pinning and versioning protocols would strengthen the methodology.

* The paper makes strong assumptions about the validity of TFDs and data quality. The proposed approach presumes clean temporal tables with unique, non-overlapping records, but real-world data often violate these constraints (e.g., acting roles, contested tenures). Robustness tests under noisy or violated TFDs, along with explicit mismatch-handling mechanisms, are needed.

* The proposed framework demonstrates limited generalization beyond structured tables. Although the SQL-based design is very elegant for relational temporal data, many time-sensitive facts originate from semi-structured or textual sources. Extending TDBench to hybrid settings that integrate temporal tables with text provenance would broaden its practical relevance.

* The analysis of RAG underperformance is insufficiently developed. While the paper notes that RAG often retrieves temporally misaligned passages (Lines 1401-1403), a more detailed investigation, such as experiments with retrieval time-filtering, temporal re-ranking, or query rewriting, could provide actionable insights for practitioners.

**Questions:**

Please address all questions raised in the weaknesses section.

---

> ### Author Response · Authors · 2025-11-21
>
> To Reviewer fr2Z (Response 1/2),
>
> We appreciate your constructive review. We respond to each of your points below.
>
> > **[W1]** *The paper lacks clarity in how “current” time is grounded. Temporal-alignment questions depend on explicit date references (e.g., “as of 2025”), yet it remains unclear how “now” is defined, stored, and exposed across datasets to ensure reproducibility and longitudinal comparability. Clearer date-pinning and versioning protocols would strengthen the methodology*.
>
> Thank you for your valuable feedback. **We do explain how the current time is grounded in Sec. B.1**, but we found that a reference to this section was missing in the main text.
>
> To briefly mention here, **you are correct that generating temporal-alignment questions depends on how “now” is defined in the database**. In TDBench, temporal expressions are generated by translating SQL queries. Temporal alignment questions – including "currently" or "as of 2025" to evaluate up-to-date knowledge – are generated by translating "now queries," which retrieve data relevant to the present moment. These "now queries" can represent “now” in various ways [1], such as checking end dates against maximum dates (e.g., 99-99-99) or NULL values depending on how data is stored. We have chosen the NULL format as it performs better in SQL-to-text translation. We provide an example here (also provided in Table 13).
>
> **SQL:**
> ```
> SELECT name, start
> FROM Leader
> WHERE Country='Netherlands' AND Role='King' AND End IS NULL
> ```
>
> **Generated questions:**
> ```
> “Who is currently serving as the King of the Netherlands?”,
> “What is the name of the individual who is the reigning King in the Netherlands?”...
> ```
> **We added a clear reference to this section in our revision (Sec 3.2, highlighted in blue).**
>
> &nbsp;
>
> > **[W2]** *The paper makes strong assumptions about the validity of TFDs and data quality. The proposed approach presumes clean temporal tables with unique, non-overlapping records, but real-world data often violate these constraints (e.g., acting roles, contested tenures). Robustness tests under noisy or violated TFDs, along with explicit mismatch-handling mechanisms, are needed.*
>
> Thank you for raising this meaningful point. To fully address it, we would like to discuss how (1) invalid TFDs and (2) data quality issues affect TDBench below.
>
> **(1) Invalid TFDs**
>
> **TDBench can exhibit robustness – generating QA pairs still aligned with the database - to invalid TFDs due to its SQL-based grounding.** For example, let us assume overlapping records where two individuals currently serve as King in the Netherlands (e.g., during a transitional period), violating the TFD $country, role \overset{T}{\rightarrow} name$. Even in this case, the "now query" from W1 successfully retrieves both records:
>
> **SQL:**
> ```
> SELECT name, start
> FROM Leader
> WHERE Country='Netherlands' AND Role='King' AND End IS NULL
> ```
>
> **Generated questions:**
> ```
> “Who is currently serving as the King of the Netherlands?”...
> ```
> **Answers via SQL:**
> ```
> {Country: Netherlands, Role=King, Name=n1, start=s1, end=NULL}
> {Country: Netherlands, Role=King, Name=n2, start=s2, end=NULL}
> ```
> Thus, while the invalid TFDs compromise our original intention on unique-answer questions, the generated QA pairs remain valid because they still accurately reflect the database content.
> That said, in complex scenarios such as multi-table joins for multi-hop questions, invalid TFDs could cause issues like incorrect TFD inference. **Following your valuable suggestion, we will additionally implement automatic TFD validation code as part of our open-source GitHub repository** to help users detect and handle such cases.
>
>
> **(2) Data quality**
>
> In contrast, data quality issues – such as factual inconsistencies and domain-specific anomalies [2]  – can affect TDBench and **lead to noisy QA pairs, which is a challenge common to all data-centric benchmarking approaches [3, 4, 5, 6, 7]**. While there is extensive work on enhancing data quality [8, 9, 10], our focus is orthogonal, where we enhance the downstream pipeline to best leverage the available data. Thus, while TDBench provides clear benefits in this context – providing a generalizable and effective tool for converting one’s data into a benchmark – we agree that integration with existing data cleaning techniques [9, 10] would further strengthen our contribution.
>
> Reflecting your valuable feedback, **we have added this discussion on invalid TFDs and data quality in our revision (Sec. 7 and Sec. B.2, highlighted in blue)**.
>
> &nbsp;

---

> ### Author Response · Authors · 2025-11-21
>
> To Reviewer fr2Z (Response 2/2),
>
> > **[W3]** *The proposed framework demonstrates limited generalization beyond structured tables. Although the SQL-based design is very elegant for relational temporal data, many time-sensitive facts originate from semi-structured or textual sources. Extending TDBench to hybrid settings that integrate temporal tables with text provenance would broaden its practical relevance*.
>
> We are grateful to see your comment on our SQL-based design. We agree that many time-sensitive facts can come from semi-structured or textual sources – particularly Wikidata and Wikipedia – which is why most existing temporal benchmarks focus on these data formats [3, 4, 5, 6, 7, 11, 12].
>
> However, **this trend actually motivates TDBench's positioning, which highlights its complementary value to current TSQA research**. Benchmarks focusing on structured temporal data remain remarkably scarce, despite the prevalence of such data in many critical domains:
> - **Healthcare**: Electronic Health Records (EHRs), patient timelines, treatment records [13, 14]
> - **Finance**: Trading records, transaction histories, regulatory data [15, 16]
> - **Enterprise**: Supply chain, Customer Relationship Management (CRM), Human Resources (HR) databases [17]
>
> By integrating database theory such as TFDs, TDBench achieves generalization to any of the above domains, enabling the use of domain-specific datasets beyond Wikipedia/Wikidata format – which is an underexplored, but vital setting.
>
> We also agree that **extending to hybrid settings (tables + text) is a valuable future direction, where some semi-structured data can be readily integrated** by transforming it into structured formats, for example, using techniques like schema inference and normalization [18, 19]. Reflecting your suggestion, **we added this discussion as a potential extension in our revision (Sec. B.2, highlighted in blue)**.
>
> &nbsp;
>
> > **[W4]** *The analysis of RAG underperformance is insufficiently developed. While the paper notes that RAG often retrieves temporally misaligned passages (Lines 1401-1403), a more detailed investigation, such as experiments with retrieval time-filtering, temporal re-ranking, or query rewriting, could provide actionable insights for practitioners*.
>
> We value your perspective. We respectfully note that **the primary goal of our analysis was to demonstrate TDBench's broad evaluation capabilities across diverse dimensions** – including cardinality (multiple/no-answer), reasoning complexity (single-hop/multi-hop), data types (real/synthetic), and compatibility with various methods (RAG, CoT, few-shot) – rather than exhaustively exploring individual techniques like RAG.
>
> That said, we agree that TDBench can aid a deeper investigation in temporal RAG setups [20]. For example, one can leverage one of TDBench’s unique strengths – 13 diverse temporal expression types – to perform robustness tests of RAG strategies across temporal expressions. **We conducted an additional analysis to exemplify such an investigation**, measuring RAG's success rate in retrieving gold documents (those containing the answer):
>
>
> | Performance Level| Temporal Relation (Success rate)|
> |---|--|
> | High (> 0.45) | after (0.67), equal (0.50)|
> | Medium (0.45–0.25) | contain (0.33), started-by (0.33), overlapped-by (0.31), start (0.31), finished-by (0.27) |
> | Low(< 0.25)    | before (0.20), met-by (0.17), meet (0.17), finish (0.15), during (0.14), overlap (0.14) |
>
> Here, the key insights are:
> - **Robustness gap across temporal expressions**: RAG shows 2-4× lower success rates on nuanced operations (e.g., "overlap": 0.14, "meet": 0.17) compared to common operations (e.g., "equal": 0.50, "after": 0.67).
> - **Performance asymmetry between inverse operations**: Performance can vary dramatically for semantically opposite pairs—"before" (0.20) underperforms "after" (0.67) by 3.3×, while some other pairs like "start" (0.31) and "started-by" (0.33) remain comparable.
>
> **We believe this kind of analysis can be further applied to diagnose diverse RAG strategies** –  such as temporal re-ranking, time-filtered retrieval, or query rewriting you mentioned. Based on your feedback, **we included this additional analysis and discussion on RAG setups (Sec. D.4, highlighted in blue)**.
>
> &nbsp;
>
> We hope our response addresses your questions. We also do appreciate your acknowledgement of our positioning, framework design, and the necessity of the time accuracy metric. Please let us know if you have any remaining questions.

---

> ### Author Response · Authors · 2025-11-21
>
> > **References**
>
> [1] Snodgrass et al., Developing time-oriented database applications in SQL, Morgan Kaufmann Publishers Inc. 1999.
> [2] Wang et al., Beyond accuracy: What data quality means to data consumers, JISTOR 1996.
> [3] Jia et al., Tempquestions: A benchmark for temporal question answering, WWW 2018.
> [4] Jang et al., Temporalwiki: A lifelong benchmark for training and evaluating ever-evolving language models, EMNLP 2022.
> [5] Gupta et al., Temptabqa: Temporal question answering for semi-structured tables, EMNLP 2023.
> [6] Kim et al., Carpe diem: On the evaluation of world knowledge in lifelong language models, NAACL 2024.
> [7] Wu et al., AntiLeakBench: Preventing data contamination by automatically constructing benchmarks with updated real-world knowledge, ACL 2025.
> [8] Batini et al., Methodologies for data quality assessment and improvement, ACL CSUR 2009.
> [9] Chu et al., Data cleaning: Overview and emerging challenges, ACM Books 2016.
> [10] Rekatsinas et al., Holoclean: Holistic data repairs with probabilistic inference, VLDB 2017.
> [11] Chen et al., A dataset for answering time-sensitive questions, NeurIPS 2021.
> [12] Tan et al., Towards benchmarking and improving the temporal reasoning capability of large language models, ACL 2023.
> [13] Johnson et al., MIMIC-III, a freely accessible critical care database, Nature Scientific Data 2016.
> [14] Pollard et al., The eICU Collaborative Research Database, a freely available multi-center database for critical care research, Nature Scientific data 2018.
> [15] Zheng et al., Global table extractor (gte): A framework for joint table identification and cell structure recognition using visual context, WACV 2021.
> [16] Zhu et al., TAT-QA: A question answering benchmark on a hybrid of tabular and textual content in finance, ACL 2021.
> [17] Jensen et al., Temporal data management, IEEE TKDE 1999.
> [18] DiScala et al., Automatic generation of normalized relational schemas from nested key-value data, SIGMOD 2016.
> [19] Baazizi et al., Parametric schema inference for massive JSON datasets, VLDB 2019.
> [20] Piryani et al., It’s high time: A survey of temporal information retrieval and question answering, arXiv 2025.

---

### Author Response · Authors · 2025-11-30
**Rebuttal Summary for New AC**

Dear Area Chair,

We sincerely appreciate the valuable efforts to keep the reviewing process fair. We summarize the rebuttal until now as follows.

&nbsp;

Overall, we had two positive score updates:
- before discussion: **6 / 4 / 4 / 6**
- after discussion: **6 / 6 / 6 / 6**

&nbsp;

**Two reviewers who initially gave a score of 4 to the paper increased their scores** after discussion:
- **Reviewer AktZ (4→6)**: We clarified how SQL-based grounding in TDBench reduces label noise, mitigates benchmark bias, and supports extensions to finer time granularities. We also demonstrated the minimal impact of translator models in QA generation and added the requested citations.
- **Reviewer YGnM (4→6)**: We clarified TDBench’s contribution to current TSQA research by substantially broadening the scope of usable temporal datasets and providing a benchmarking framework that can be easily updated as data evolves. We also discussed the value of the proposed time accuracy metric, which gives insights such as performance overestimation that was not evident in answer-only evaluations.


We note that **the score increases occurred before the leak incident (11/27)**, and **both Reviewers clearly explained why they raised their scores** through official comments:
- **Reviewer AktZ (11/26)**: *“I thank the authors for their response. They have addressed my concerns, and I have updated my review score accordingly.”* (4→6)
- **Reviewer YGnM (11/24)**: *“Thanks for your detailed response. The additional insights you provided have successfully addressed my concerns. As a result, I am happy to raise my score for the submission.”* (4→6)

&nbsp;

We also addressed the concerns of the two reviewers who initially gave a score of 6:
- **Reviewer fr2Z**: We clarified how temporal alignment questions are generated and how the underlying data quality issues affect TDBench. We also discussed how TDBench can be extended to incorporate semi-structured data and aid investigation in advanced temporal RAG settings.
- **Reviewer wK6u**: We clarified the widespread availability of temporal databases and the scope of TDBench’s automation in leveraging these existing data. We also discussed how SQL-based grounding lowers hallucination risks compared to LLM-only QA generation methods and clarified the rationale for date information prompting to ensure fair and consistent comparisons as a benchmark paper.

We unfortunately did not receive follow-up feedback from these reviewers.

&nbsp;

We would be grateful if you could consider this rebuttal process in your final decision. We respectfully note that **the paper revisions and updated reviewer scores were made solely through constructive dialogue**, which we believe substantially strengthened the clarity and contributions of our paper.


We again thank the ACs and all reviewers for their thoughtful engagement throughout the rebuttal process. We hope this summary assists with the remainder of the process.

&nbsp;

Best regards,
Authors of paper #16700

---

### Meta-Review · Area_Chair_Ao8m · 2026-01-07

**Summary:**

The paper presents a new QA dataset, named TDBench, that is automatically constructed to handle time-sensitive factual questions. The dataset spans general and domain-specific datasets and constructed from database entries. The proposed dataset covers a diverse domains, going beyond Wikipedia and more domain-specific knowledge sources (e.g., Kaggle, Legal database, etc). Furthermore, the dataset covers diverse temporal relations (overlap, contain, start by, end by, etc). The reviewers acknowledge the proposed datasets value in comparison to existing body of temporal QA benchmark, and the paper is clearly written with rich analysis. The dataset is well constructed and documented. While there's some limitations such as only covering questions that can be answered solely from databases (rather than combination of text + DB), overall the paper presents a valuable resources and I recommend acceptance.

**Reviewer Concerns:**

think some of the reviewer concerns (e.g., not covering text + database from Reviewer fr2Z, not covering open/uncertain intervals or requires granularities beyond dates from Reviewer AktZ, relying on high-quality DBs from Reviewer wK6u) are not addressed by authors, as these are real weaknesses of the paper/approach.

**Reviewer Scores:**

AktZ and YGnM said they'd increase scores, so I assume they would have.

---

### Decision · Program_Chairs · 2026-01-26

Accept (Poster)